# The Management of Railway Operations during the Planned Interruption of Railway Infrastructure

**Zdenka Bulková** *[ID], **Jozef Gašparík** [ID] **and Vladislav Zitrický** [ID]

Department of Railway Transport, Faculty of Operation and Economics of Transport and Communication, University of Žilina, Univerzitná 8215/1, 01026 Žilina, Slovakia; jozef.gasparik@uniza.sk (J.G.); vladislav.zitricky@uniza.sk (V.Z.)
* Correspondence: zdenka.bulkova@uniza.sk; Tel.: +421-41-513-34-11

**Abstract:** A planned interruption of railway infrastructure is a situation where the operation of the track line or the operation of railway transport is limited. If there is also a restriction on the railway infrastructure, it means there will be complications not only for passengers but, above all, for railway undertakings operating freight transport. However, because of the planned railway infrastructure interruption, the quality of services provided not only to passengers but also to freight transport is decreasing. The aim of this paper is to propose effective planned maintenance works based on the analysis and evaluation of the processes performed during the planned railway infrastructure interruption or restriction. The research describes the process of affected railway infrastructure from technical, cost, and safety points of view. A methodological procedure is proposed under the condition of the Czech infrastructure manager. The main method is the calculation of the costs for the railway infrastructure manager and railway operator during the infrastructure interruption. The application part is undertaken using two interrupted lines according to the established alternative timetable in the area of České Budějovice.

**Keywords:** railway infrastructure; railway transport; planned railway interruption; railway operation; infrastructure manager costs; operator costs

## 1. Introduction

Railway infrastructure plays a key role in modern transport systems, which require constant innovation and precise management and maintenance [1]. The basic goals of the infrastructure manager are to ensure regular, reliable, and safe railway infrastructure and to provide it to operators of railway passenger and freight transport in a non-discriminatory manner [2]. Since the property components of rail transport, especially rail infrastructure, are capital-intensive and have a long life cycle, their operation and maintenance require a long-term and sustainable strategy. Strategic planning involves gathering information, setting goals, translating goals into specific goals, and taking action to achieve those goals [3]. In the transport policy of the European Union, the ownership model of the railway infrastructure is preferred to the vertically controlled model. In this system, one entity manages and takes care of the infrastructure, while transport services on this network are then provided by one or more operators (rail operators) who compete (unbundling) [4]. It focuses on increasing competitiveness in rail transport and making investments in rail and road infrastructure at a uniform level.

Infrastructure managers (hereafter IMs) are either the owners of the railway infrastructure or companies that have obtained concession contracts. They are responsible for the safety and maintenance of railway lines but mainly for enabling the various train operators to access the line. According to [5], in all European Union countries, IMs are national monopolies, most of which are publicly owned. They mention a few exceptions, such as some German railway lines or the Perpignan–Figuears line. IMs have far-reaching effects

on the business models of railways and industry players, resulting in them being highly regulated.

It is similar in the Czech Republic, where the state is the owner of the national railway and most of the regional railways. They manage this state property, and the function of the railway owner on it is in accordance with Act 266/1994 Coll. about railways, which is carried out by the Czech infrastructure manager. According to the Act on Railways, the owner of the railway is obliged to ensure the maintenance and repair of the railway to the extent necessary for its operability. As far as the national or regional railways are concerned, the owner is also obliged to take care of the development and modernization of the railway to the extent necessary to ensure the transport needs of the state and the transport services of the regional territory [6,7]. To fulfill the letter of the law regarding infrastructure maintenance and, at the same time, maintain safety in the organization and operation of rail transport, planned interruptions of railway traffic are held. For the activities of the transport process to proceed safely and undisturbed (smoothly), it is necessary to interrupt a part of the infrastructure for a certain period. That means that there will be restrictions, which, to a lesser or greater extent, manifest themselves in the delay of passenger and freight transport [8]. Interrupting railway traffic is a method of transport and operational use of transport road equipment requiring the adoption of special technological and technical measures, during which there is a restriction on the operation of the track and possibly also on the restriction of the operation of rail transport [9]. The interruption of railway traffic is carried out by the railway operator as part of regular maintenance, repair, or reconstruction of the railway transport route, safety equipment, traction line, or preservation of a clear crossing. All this is to maintain the safety of train operations. The interruption of railway traffic is carried out due to modernization or reconstruction or due to repair or maintenance, and it must be planned in such a way that the negative impacts on railway transport are as small as possible [10]. Therefore, it is necessary to pay attention to the proper preparation and organization of rail traffic interruption and to use the necessary technologies or applications for this [11]. During repairs, reconstruction, or construction of new tracks, the normal operation of railway or station tracks, switches, and fixed electric traction equipment may be interrupted, or the safety equipment may be completely or partially switched off. The interaction between pantographs and catenaries in ensuring the uninterrupted supply of electricity to trains is described in the papers [12–14].

This paper deals with the analysis and evaluation of the processes that are carried out during the planned interruption or limitation of railway traffic. The aim of this research is to propose a new methodical procedure for evaluating variants of work organization during the interruption of railway traffic, considering the technical, cost, and safety aspects based on knowledge of the processes of the affected railway operation during the infrastructure interruption. The proposed methodological procedure is verified using the example of the infrastructure of the Czech infrastructure manager, which is currently the Správa železnic. The proposal uses four types of methods that relate to the planned interruption of railway traffic. The methodology also focuses on the determination of technological procedures and safety in the implementation of the interruption of railway traffic. The ambition of the proposed solution is to expand the hitherto applied approaches to the solution of the mentioned issue and provide a global view to all actors implementing rail transport, i.e., to the managers of the railway infrastructure and operators, especially in passenger transport. The goal is to determine, in an objective manner, the most advantageous option for interrupting railway operations during maintenance. The application part of this research is carried out on two interrupted lines according to the established alternative timetable.

## 2. Literature Overview

The rail industry is currently investigating viable measures to strengthen the track and reduce the need for frequent maintenance. These remedial measures should not only be cost-effective but should also be consistent with the sustainability of the railway

infrastructure, considering growing environmental concerns. The rail system is undeniably vital for the transport of passengers and goods along the main rail corridors. To meet the growing demand for freight and public transport, the railway industry is constantly using advanced technologies to model infrastructure capacity [15]. This led to the development of faster passenger trains and increased capacity for freight trains. Increased train speed and higher tonnage in recent years have escalated the deterioration of the railway infrastructure. As a result, frequent maintenance or even reconstruction became necessary [16].

The issue of the interruption of railway traffic is very extensive, from planning through the correct process to the economic and technological consequences. The authors of [17] describe the distribution, technology, and specific examples of the interruption of railway traffic on railway lines in the Czech Republic. The authors solve the problem of optimizing the technology and organization of the interruption of railway traffic and the consequences for passengers. Passenger transfers are very problematic (at stations from/to connecting trains). General transportation costs are rising. A complication is also the occurrence of several parallel interruptions of railway traffic in one specific region (for example, two railway lines next to each other). The implementation of freight transport is also very important because the diversion of trains is not always possible.

Increasing supply in railway networks comes at the cost of an increased need for infrastructure maintenance. It also means adjusting the schedule due to the long maintenance or property of the building. In the research of [18], the train timetable adjustment problem (TTAP) was solved for a given station and free track, which led to an alternative timetable that minimized the deviation from the original timetable. To solve TTAP, the authors propose a mixed-integer linear programming (MILP) model and apply retiming, reordering, truncation, and cancelation to generate alternative timetables. The model presents an extended periodic event scheduling problem (PESP) formulation and introduces new constraints for the cancelation and rescheduling of train lines, while short detours are applied in the preprocessing step. In addition, with some modifications, it can be used for interrupted management. Operators and infrastructure managers could use it to automatically generate optimal alternative timetables at a macroscopic level in case of maintenance or construction work, thus coordinating transport for the entire network.

The author of [19] presented a survey on possible maintenance activities, associated planning problems, and mathematical models developed for these problems. Planning infrastructure maintenance activities is important at all levels of planning—strategic, tactical, and operational. Depending on the time of planning, assets can be categorized as preventive or corrective. The author focuses on preventive measures, which are defined as maintenance that can be planned long in advance, such as the renewal and replacement of existing tracks. The author also distinguished two types of maintenance: the main ones, which cause conflicts with planned train routes, and smaller ones that do not interfere with train operation. Additionally, a train route involves the infrastructure capacity required to run a train between two locations during a given period [20].

Maintenance can be scheduled for a different time of day or even on other days. In another paper [21], these cases were divided into three main categories: overnight asset component maintenance, weekend asset component maintenance, and daytime asset component maintenance. From a traffic point of view, night maintenance is the most desirable because it would cause the least disturbance to the operation or not disturb the operation at all. However, night maintenance is often not favorable for employees, or the given time is insufficient. Longer maintenance will obviously interfere with traffic during the day. Examples of such ownership are renovation works on a station platform that can cause the occupation of adjacent tracks for up to several weeks or the repair of signals along a track between two stations lasting a full working day. As the author considers major maintenance, adjustments to the timetable are necessary. Three approaches to solving the major maintenance problem are distinguished: (a) planning maintenance windows independently of the timetable, which can also be considered a strategic problem;

(b) adjusting timetables for a given asset component maintenance; and (c) planning train traffic and simultaneously monitoring the maintenance of the asset component.

The authors of [22] assessed and dimensioned the maintenance windows before creating a timetable for the interruption of railway traffic. Providing a certain pattern of maintenance windows without trains would have an impact on overall maintenance costs and would also affect subsequent timetable development. The authors tried to distribute the departures evenly, but due to predetermined maintenance intervals, the model had to cancel train routes, adjust event times (retiming), or extend journey times. A cost model was compiled to establish dimensions for the time windows and assess their resulting maintenance costs as well as passenger and freight transport costs. In the paper of [23], the authors focused on the interruption of railway traffic to the existing timetable by adjusting train traffic and interruption simultaneously. For this purpose, they proposed an expert rule-based local search heuristic that uses the Problem Search Space approach. As a benchmark, they simulated the practice of manual planning. Their heuristics reduced total and maximum train delays by 17–34%. No comparison with the optimal solution is given.

In the case of conventional railway lines, when breakdowns occur, dispatchers have the difficult task of quickly finding feasible rescheduling solutions to restore normal conditions as soon as possible. Such management of disturbances in railway passenger transport was dealt with by [24]. The authors of [25] propose a model system for railway system management that combines a microscopic simulation model with a matching tool capable of considering passenger flows on the network. An application on a real regional track in Campania (Italy) shows the benefits of the proposed approach for performing offline analyses of intervention solutions and helping dispatchers make decisions during critical events to improve service quality. The authors of [26] present a model that optimizes the production schedule for both trains and preventive maintenance. Scheduled maintenance activities cannot be canceled but can be moved within pre-defined time windows. Trains can be shifted in time, diverted to other parts of the geography, or even canceled. The goal of optimization is to find the best possible traffic flow, given a fixed set of planned maintenance activities. The authors of [27–29] researched the modeling of train routes and the modification of timetables for given maintenance. From the user's point of view, one of the main strengths of rail transport systems is reliability, that is, the ability to respect timetables and travel times. However, interruptions to scheduled services can occur for several reasons (stochastic fluctuations in travel demand, infrastructure damage, convoy breakdowns, etc.), causing a significant reduction in service levels. One of the objectives of the railway network operator is, therefore, to minimize the duration of such interruptions and their negative effects. This issue, from the point of view of passenger satisfaction, was dealt with by [30–33].

## 3. Theoretical Backgrounds

In the EU, in the vertically separated model of the railway market, the railway is usually owned by the state, which is most often represented by the Ministry of Transport. The owner of the railway is obliged to permanently maintain the railway in operable condition, to restore the operation of the railway after an accident or after an extraordinary event, and to take care of the maintenance and development of the railway in accordance with the safety and flow of traffic. The duties of the railway infrastructure manager (IM) primarily include the fair and non-discriminatory allocation of railway infrastructure capacity to licensed railway undertakings [34]. In cases of operational emergencies, the infrastructure manager is obliged to find out the cause of the accident or extraordinary event on the track and report it to the investigating body. In the research, the IM in the Czech Republic was selected, which is the entity Správa železnic. The Czech infrastructure manager issues and constantly updates the internal regulations by which the employees are guided both during normal operation and, above all, during various emergencies to which the interruption of railway traffic belongs. These regulations are also binding for operators that operate rail transport on railways, as well as foreign legal entities working

on railway facilities. The interruption of railway traffic on national and regional railways is supported by Act No. 266/1994 Coll. on Railways [6], and its organization is dealt with by internal regulation D7/2 [35]. It is also necessary to follow the regulations on the traffic and signaling system D1 [36], the regulation on simplified traffic control D3 [37], the regulation on traffic control on tracks equipped with radio blocks D4 [38], the regulation on works on railway superstructure S3/1 [39], and, last but not least, safety and health protection at work, which are addressed by regulation Bp1 [40]. There are 9349 km of railway lines in the Czech Republic, of which 7279 km are single-track, 2070 km are double-track and multi-track, and 3258 km are electrified lines of alternating current and one-way traction systems. One hundred twenty-six operators operate rail transport on these lines [41]. The number of train paths in the timetables in 2022 and 2023 by category in the Czech Republic is shown in Table 1. The recalculation of allocated train paths per 100 km of lines, which on one of the densest networks in the EU reaches an average of 176, is supplemented.

**Table 1.** Number of train paths in timetables in 2022 and 2023 (authors, according to [41]).

| Indicator | 2022 | 2023 |
|---|---|---|
| Regional trains (Reg) | 8842 | 10,574 |
| Fast trains (F) | 1098 | 1473 |
| Higher quality trains (EC, EN, Ex, IC, LE, SC, Railjet) | 264 | 347 |
| Freight express (Fex) | 471 | 703 |
| Freight running trains (Fr) | 766 | 899 |
| Freight commuter trains (Fc) | 1278 | 926 |
| Locomotive trains (Loc) | 626 | 557 |
| The number of allocated train routes on the network | 15,479 | 17,560 |
| The length of the tracks of the Czech infrastructure manager network | 9358 | 9355 |

Transport performance in train kilometers is affected by the interruption of railway traffic. During the continuous interruption of railway traffic due to the diversion routes, transport performance is increased in railway freight transport and, in some cases, also in railway passenger transport. In railway passenger transport, there may be a decrease in transport performance if alternative bus transport is used instead of a train on the railway transport route. Figure 1 shows an overview of the transport performance of operators in passenger and freight transport on the railway network in the Czech Republic in the years 2019–2023, which is expressed in millions of train kilometers.

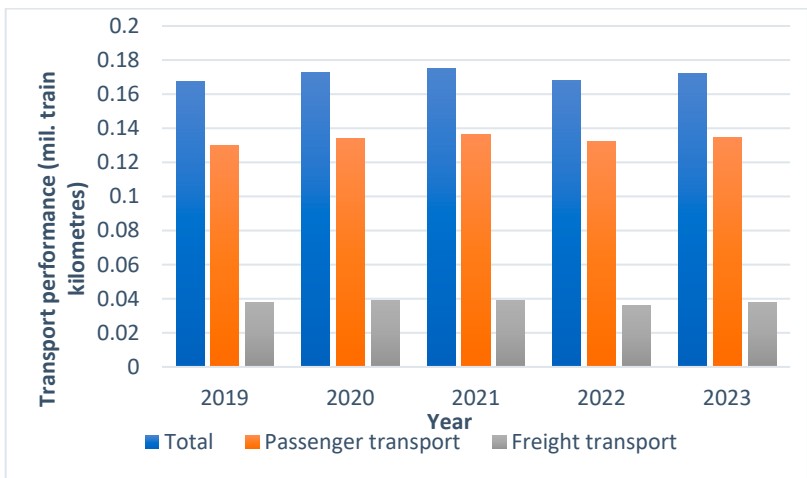

**Figure 1.** Performance of operators on the Czech infrastructure manager network in 2019–2023 in millions of train kilometers (authors, according to [41]).

Table 2 shows selected indicators associated with the interruption of railway operations in the Czech Republic and the number of repaired kilometers, tracks, switches, and other components for the years 2022 and 2023 that are necessary to ensure operability. Based on the authors' calculation, maintenance and, therefore, closures affected an average of 25% of the length of the railway network per year (considering that some maintenance activities were carried out simultaneously on the same section of infrastructure).

**Table 2.** Selected activities related to railway traffic interruption in the period 2022–2023 (authors, according to [41]).

| Indicator | Measurement Unit | Quantity | |
|---|---|---|---|
| | | **2022** | **2023** |
| Adjustment of the geometric position of switches | pcs | 717 | 1294 |
| Cleaning—switches | pcs | 137 | 205 |
| Routine inspection and welding—switches | pcs | 305 | 359 |
| Exchange of sleepers | pcs | 172,121 | 315,446 |
| Adjusting the geometric position of the tracks | km | 1256 | 1658 |
| Cleaning—tracks | km | 82 | 206 |
| Routine inspection and welding—rails | km | 221 | 662 |
| Rail replacement | km | 293 | 512 |

According to [42,43], there are two types of such interruptions within the planned interruption of railway traffic: planned interruption of railway traffic (as part of at least a weekly plan) and unplanned interruption of railway traffic (not included in the weekly plan). The interruption of railway traffic is carried out on different timelines. Operative interruption of railway traffic takes place within a predetermined time range, number, and duration of interruption of the traffic. Operative interruption of traffic can only be used in exceptional cases. Infrastructure interruptions take place in different timelines. According to the duration of the infrastructure interruption and the time in which they take place, infrastructure interruptions are divided into day, night, and continuous. Table 3 shows the types of infrastructure interruptions according to their duration and the time in which they take place.

**Table 3.** Types of railway infrastructure interruptions.

| | Type of Interruption of Railway Traffic |
|---|---|
| According to its duration | Per day—day of infrastructure interruption<br>Per night (6.00 P.M.–6.00 A.M.)<br>Continuously—continuous mode, which exceeds the parameters of night infrastructure interruption |
| According to the range of interrupted devices | Complete interruption of service (alternative bus transport or diversion required)<br>Partial restriction of the railway traffic |

Railway operations may be interrupted due to the maintenance of the track, traction line, or safety equipment. Considering the slope ratios and the highest permitted speed in the given section, it is possible to cross the so-called road by driving with the pantographs retracted. If the slope conditions are not favorable, the operator must implement measures. One of them is that it will replace dependent traction-drive vehicles with independent traction-drive vehicles. With this measure, however, it depends on the number of trains for which the compensation should take place and whether the operator is able to make the compensation in such a quantity [44]. Within railway stations, operations may be interrupted in the entire station or only in some of its parts. It is possible to interrupt operation on individual traffic tracks or a group of traffic tracks. An example of such an

interruption is shown in Figure 2. In these cases, it is possible to operate railway traffic on the remaining traffic tracks.

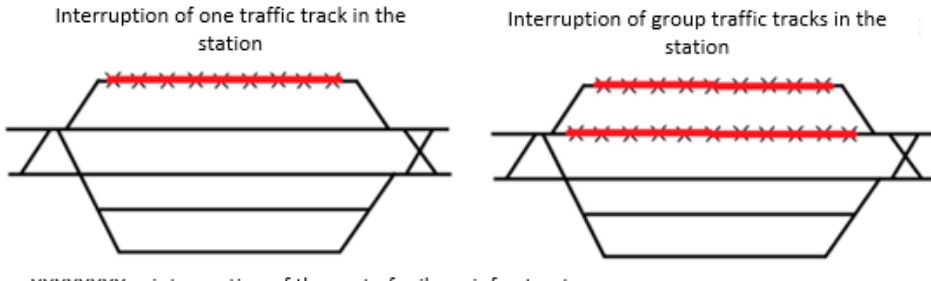

**Figure 2.** Example of interruption of one transport track and a group of station transport tracks [45].

Another interruption in the station may be the switch, which is shown in Figure 3. Maintenance work in these parts of the station is not directly concerned with the complete interruption of the station tracks. But it is always related to their partial limitations and use. It is not possible to carry out departures or arrivals of trains in all directions, and in the case of an interruption at the switch, it is not possible to carry out the circulation of sets by locomotives.

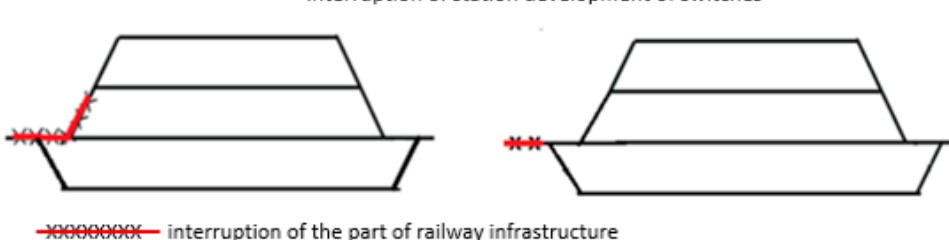

**Figure 3.** Interruption of the traffic of the station switch [45].

In the case of multi-track sections, the rule is to maintain single-track operation. To maintain the flow of rail traffic, Figure 4 shows the interruption of the track on a single-track line and a double-track line.

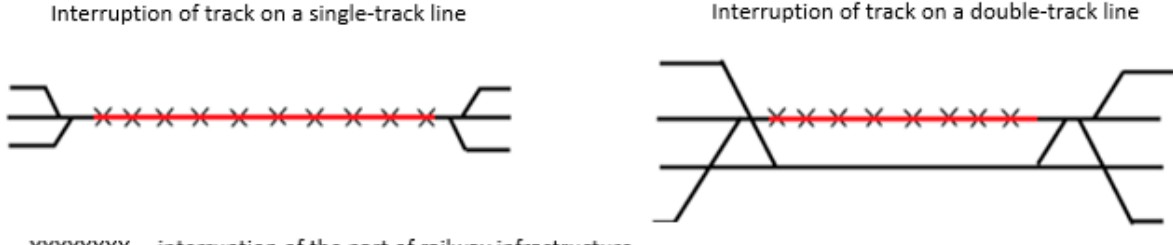

**Figure 4.** Interruption of track on single-track and double-track lines [45].

Such an interruption of traffic on a single-track line is classified as an interruption with a complete stop of railway traffic. However, there are sections where there is a line without detour that can be used for running trains as part of the traffic measure. This situation is shown in Figure 5.

However, there are sections where there is a detour route that can be used for running trains as part of the traffic measure. The track with a detour route is shown in Figure 6.

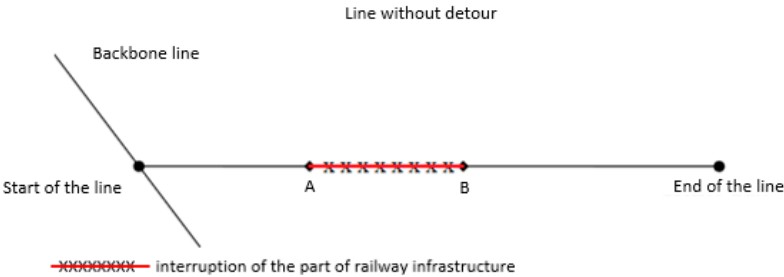

**Figure 5.** Interrupted track on a single-track line without the possibility of a detour route [45].

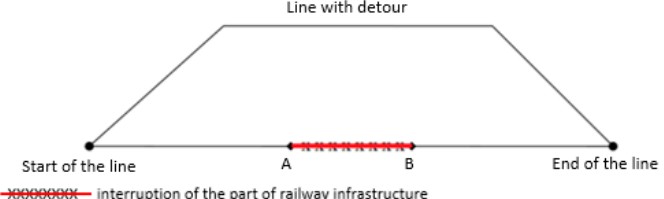

**Figure 6.** An example of a single-track line with an interrupted track with a detour route [45].

### 3.1. Planned Railway Infrastructure Interruption

Before the start of the planned interruption of railway traffic, an application must be submitted to the locally relevant customer for the interruption of railway traffic. The request is submitted by the applicant for the railway infrastructure interruption. Professional units responsible for the safe operation of parts of the railway infrastructure will draw up a plan for the railway infrastructure interruption for repair and modernization actions according to the current technical status of a specific infrastructure device. Maintenance works have a set maintenance schedule as a basis, which considers, e.g., vegetative rest. Such documents can also be simulations, for example, for modeling the throughput performance of railway traffic on a track where maintenance is taking place [46]. The railway undertaking operating passenger transport shall ensure that in the railway stations affected by the interruption of traffic, passengers are informed about alternative bus transport, train queues, and the location and marking of stations intended for boarding or output passengers [47].

There is much research on planned infrastructure interruptions that addresses the issue through operational, tactical, strategic, and simulation models. As part of our research, we focused on the intersection between tactical and strategic models (area marked in red), which includes a plan of operation from the point of view of cost, technology, and safety. Figure 7 shows the models of planned infrastructure interruption.

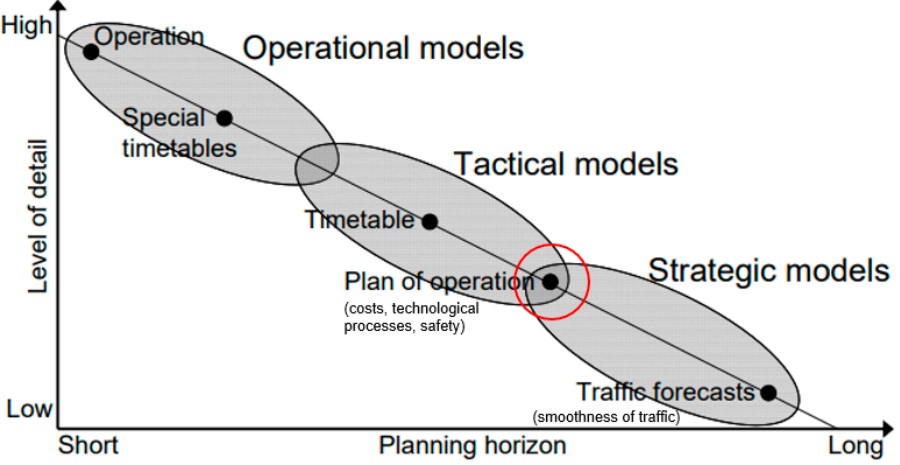

**Figure 7.** Types of models during infrastructure interruption.

The practical application of the proposed methodology will be demonstrated in the case of a planned interruption of traffic on two selected interrupted lines: **České Budějovice–České Velenice (number 705)** and **České Budějovice–Horní Dvořiště (number 706)**.

The planned suspension of operations on the České Budějovice–České Velenice line (705) was intended for repair and maintenance work on the České Velenice–České Budějovice track section. The maintenance was divided into five stages, A–E (Table 4), which took place every two working days. In each stage, the interstation section between two neighboring stations was interrupted by track, and at the same time, the voltage was interrupted in the same section with a continuation up to the České Velenice railway station. As part of the operation of railway transport on both lines, all passenger trains were replaced by alternative bus transport for the duration of the infrastructure interruption, and diversion routes were not used. Freight trains were waiting for the end of the infrastructure interruption, and, subsequently, after the resumption of operations on these lines, passenger and freight trains were again put into normal operation. As this was a planned infrastructure interruption, freight trains did not use diversion routes and waited in stations. The locations of railway lines for the infrastructure interruption on lines 705 and 706 are shown in Figure 8, and the locations of railway lines for the stages of infrastructure interruption are shown in Figure 9.

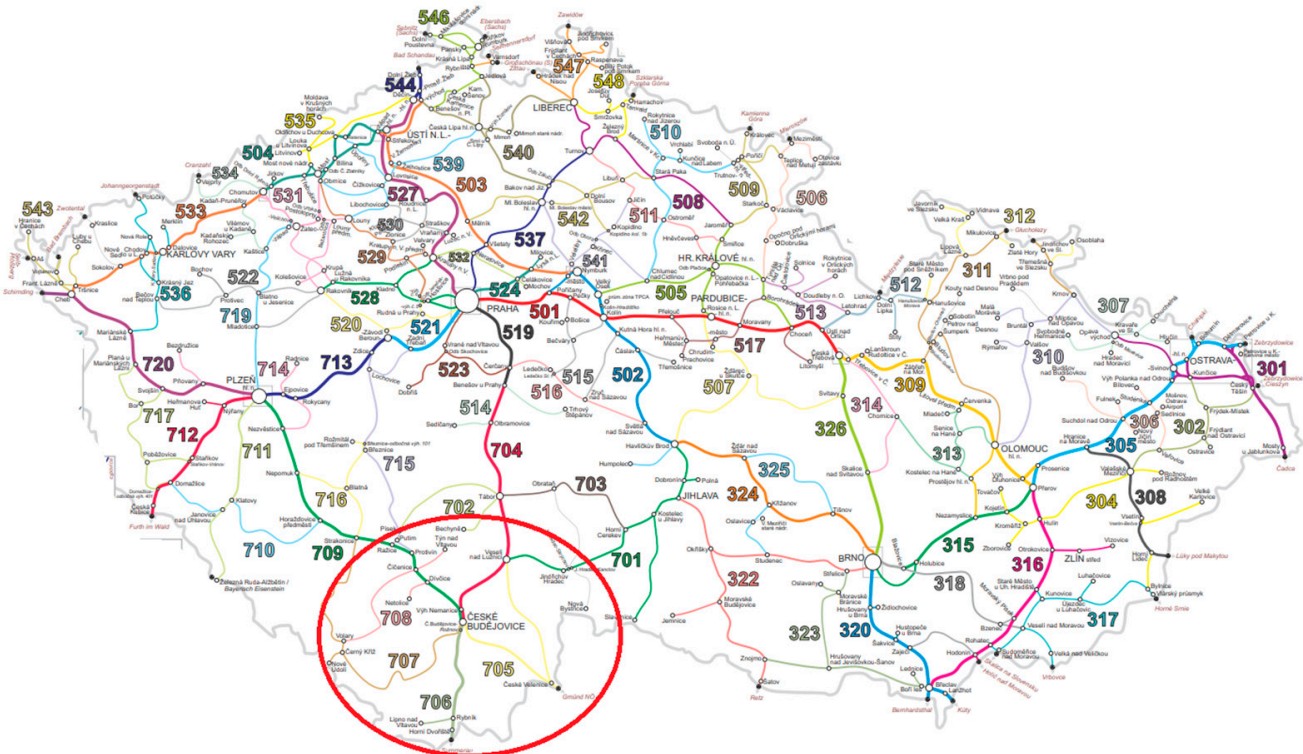

**Figure 8.** Locations of railway lines 705 and 706 within infrastructure interruption in the Czech Republic (authors, according to [48]).

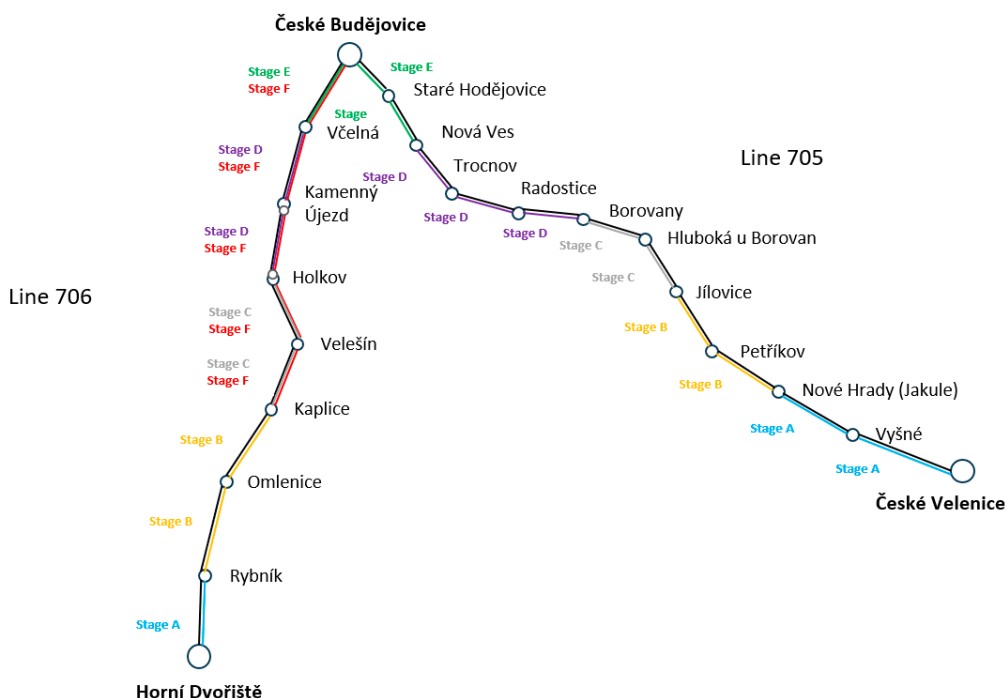

**Figure 9.** Railway lines with the stages of infrastructure interruption.

**Table 4.** Stages of maintenance activities on the České Budějovice–České Velenice line (authors, according to [49]).

| Stage | Description |
|---|---|
| Stage A | Determined range of days (date), which was always from 7:30 A.M. to 3:35 P.M.<br>The 13 km long České Velenice–Nové Hrady (Jakule) track section, which was interrupted by tracks.<br>The distance for alternative bus transport from the České Velenice station to the Nové Hrady station via the Vyšné railway stop, with the route being determined by the infrastructure interruption traffic order, was 14.7 km.<br>During maintenance, the train consists of two Regio Panter electric units. In the Nové Hrady (Jakule)–České Velenice and České Velenice–Nové Hrady (Jakule) track section, 8 regional trains were replaced by alternative bus transport.<br>Freight trains were waiting for the infrastructure interruption to end. Their operation began at 3:45 P.M. |
| Stage B | Determined range of days (date), which was always from 7:20 A.M. to 3:30 P.M.<br>The Nové Hrady (Jakule)–Jílovice track section that has been interrupted by tracks.<br>Freight trains were waiting for the infrastructure interruption to end. Their operation began at 3:40 P.M. |
| Stage C | Determined range of days (date), which was always from 7:10 A.M. to 3:20 P.M.<br>The Jílovice–Borovany track section was interrupted from the track.<br>For stages B and C, 8 regional trains in the Nové Hrady (Jakule)–Jílovice–Borovany and Borovany–Jílovice–Nové Hrady (Jakule) track section were replaced by alternative bus transport. It is a 16.4 km long track section. The distance for alternative bus transport from Nové Hrady station to Borovany station via Petříkov railway stop, Jílovice railway stop, Jílovice station, and Hluboká u Borovan railway stop, with the route being determined by infrastructure interruption traffic order, was 28.1 km.<br>Freight trains were waiting for the infrastructure interruption to end. Their operation began at 3:30 P.M. |
| Stage D | Determined range of days (date), which was always from 7:15 A.M. to 2:10 P.M.<br>The Borovany–Nová Ves track section was interrupted.<br>Freight trains were waiting for the infrastructure interruption to end. Their operation began at 2:20 P.M. |
| Stage E | Determined range of days (date), which was always from 7:20 A.M. to 2:00 P.M.<br>The Nová Ves–České Budějovice track section was interrupted.<br>For stages D and E, 6 regional trains in the České Budějovice–Nové Hrady–Borovany and Borovany–Nové Hrady–České Budějovice track section were replaced by alternative bus transport. It is a track section 19.9 km long. The distance for alternative bus transport from Borovany station to České Budějovice station going through the railway stops Radostice and Trocnov, station Nová Ves, and the railway stop Staré Hodějovice, with the route being determined by the infrastructure interruption traffic order, was 24.9 km.<br>Freight trains were waiting for the infrastructure interruption to end. Their operation began at 2:10 P.M. |

The České Budějovice–Horní Dvořiště line (706) is a single-track electrified line included in the TEN-T system. Following the European division, the routes of railway corridors were established in the Czech Republic, which is an IV. transit corridor [50], and this is the national designation of this corridor, which is the main long-distance railway connecting Stockholm–Dresden–Děčín–Prague–Tábor–Veselí and Lužnicí–České Budějovice–Horní Dvořiště–Linz–Salzburg–Ljubljana–Rijeka–Zagreb [51,52]. The planned infrastructure interruption on this line was intended for repair and maintenance work in the Horní Dvořiště track section of the state border in České Budějovice. Maintenance was divided into six stages, A–F (Table 5), which always took place on working days.

**Table 5.** Stages of maintenance activities on the České Budějovice–Horní Dvořiště line (authors, according to [49]).

| Stage | Description |
|---|---|
| Stage A | Determined range of days (date), which was always from 9:10 A.M. to 4:45 P.M. <br> The Horní Dvořiště–Rybník track section, including Horní Dvořiště station, was 7.3 km long. <br> The distance for alternative bus transport from Horní Dvořiště station to Rybník station, with the route being determined by infrastructure interruption traffic order, was 8.4 km. <br> Freight trains were waiting for the infrastructure interruption to end. Their operation began at 4:55 P.M. |
| Stage B | Determined range of days (date), which was always from 9:00 A.M. to 4:25 P.M. <br> The Rybník–Kaplice track section, including the entire Omlenice station. <br> The distance for regional trains from Loučovice station to Kaplice station (Rybník–Lipno nad Vltavou line) is a railway transport route that is 35.8 km; alternative bus transport was 44.6 km. <br> The distance for alternative bus transport of fast trains from the Rybník station to the Kaplice station is a railway transport route that is 16.7 km; alternative bus transport was 20.9 km. <br> The distance for alternative bus transport of fast trains from the Horní Dvořiště station to the Kaplice station is a railway transport route that is 24 km; alternative bus transport was 25 km. <br> Freight trains were waiting for the infrastructure interruption to end. Their running began at 4:35 P.M. |
| Stages C and F | Determined range of days (date), which was always from 8:45 A.M. to 4:15 P.M. <br> The Kaplice–Holkov track section, including the entire Velešín station. <br> The distance for regional trains from the Kaplice station to the Holkov station is a railway transport route of 12.3 km, while the alternative bus transport is 11.4 km <br> The distance for alternative bus transport of fast trains from the Omlenice station to the Kaplice station is a railway transport route that is 6.6 km; alternative bus transport was 20.1 km. <br> Freight trains were waiting for the infrastructure interruption to end. Their operation began at 4:25 P.M. |
| Stage D | Determined range of days (date), which was always from 8:50 A.M. to 4:10 P.M. <br> The Holkov–Včelná track section, including the entire station Kamenný Újezd. <br> The distance for regional trains from Holkov station to Včelná station is a railway transport route that is 12.8 km; alternative bus transport was 11 km. <br> The distance for alternative bus transport of fast trains from Horní Dvořiště station to České Budějovice station is a railway transport route that is 56.8 km; alternative bus transport was 51 km. <br> Freight trains were waiting for the infrastructure interruption to end. Their operation began at 4:20 P.M. |
| Stage E | Determined range of days (date), which was always from 8:55 A.M. to 4:00 P.M. <br> The Včelná–České Budějovice track section with a length of 8.1 km. <br> The distance for regional trains from the Včelná station to the České Budějovice station is a railway transport route that is 8.1 km; alternative bus transport was 8 km. <br> The distance for alternative bus transport of fast trains from Horní Dvořiště station to České Budějovice station is a railway transport route that is 56.8 km; alternative bus transport was 51 km. <br> Freight trains were waiting for the infrastructure interruption to end. Their operation started at 4:10 P.M. |
| Stage F | Determined range of days (date), which was always from 8:45 A.M. to 4:15 P.M. <br> The train heading toward Velešín was interrupted, along with switch no. 12 in Kaplice station. <br> Freight trains were waiting for the infrastructure interruption to end. Their operation began at 4:25 P.M. |

### 3.2. Analysis of Costs for Activities Related to the Railway Infrastructure Interruption

The costs incurred by the Czech infrastructure manager in the years 2010–2019 to ensure the operability of the railway as part of repairs and maintenance on the entire network are shown in Figure 10 and are calculated per one kilometer of track.

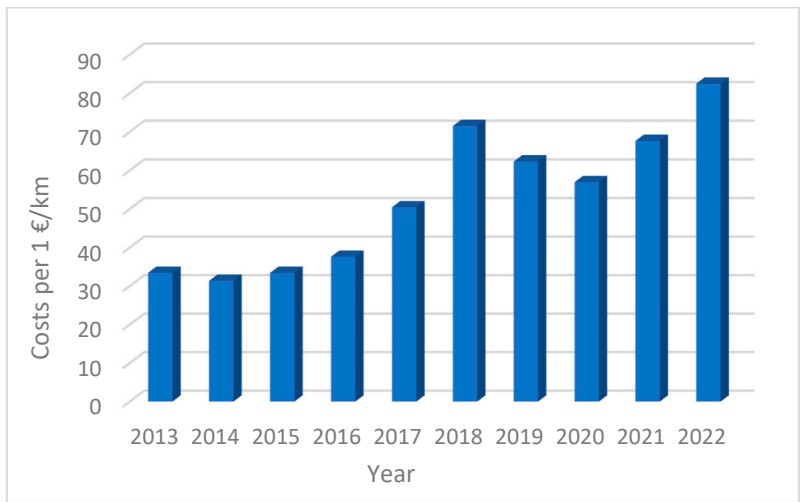

**Figure 10.** Unit costs spent on the operability of the infrastructure of the Czech infrastructure manager [EUR/km] (authors, according to [49]).

The highest-cost item is repair and maintenance work. Figure 11 shows the unit sales for the sale of services converted to EUR 1/km. The line graph shows how much of the total revenue is made from the use of the railway infrastructure for passenger and freight transport and the allocated capacity converted to EUR 1/km.

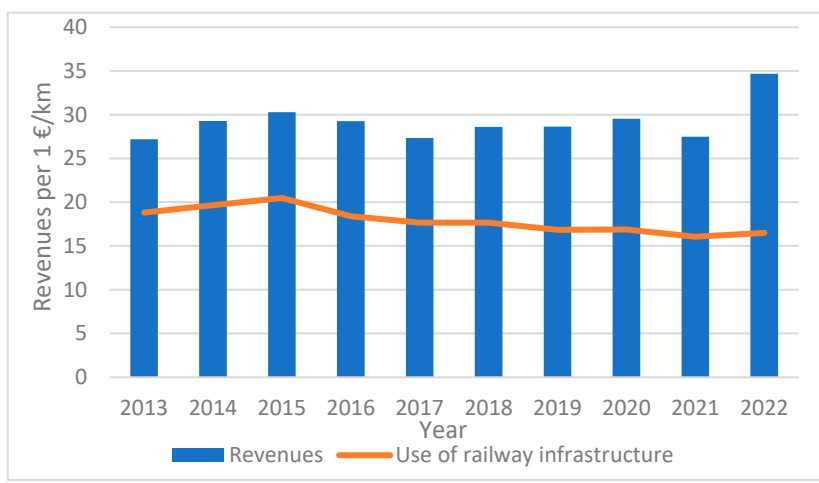

**Figure 11.** Revenues from the sale of products and services of the Czech infrastructure manager [EUR/km] (authors, according to [49]).

The total price for the use of a track by a train operated by the Czech infrastructure manager depends on the length and parameters of the track traveled, the train parameters, the basic price, and the application of the product factor and specific factors. The price is determined by a calculation that is based on the actual scope of performance of the operators and the train kilometers traveled in the given billing period on the railway network of the Czech infrastructure manager.

### 3.3. Safety during the Interruption of Railway Traffic

Ensuring safe and smooth operation is a priority in the operation of railways and railway transport. In the case of an emergency, the risk of injury or accident is higher. The number of all accidents in the years 2013–2022, as well as injuries that occurred as part of maintenance work during the interruption of traffic, is shown in Figure 12.

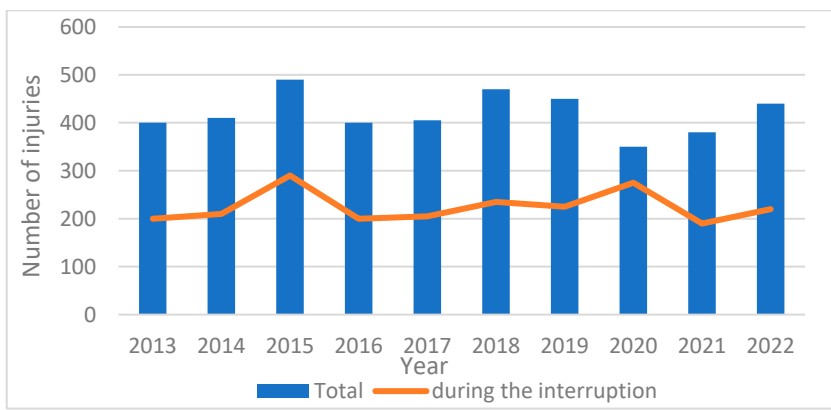

**Figure 12.** The number of injuries to employees of the Czech infrastructure manager in 2013–2022. (authors, according to [49]).

## 4. Methodology

The proposal for the methodology of the management of works during the interruption of railway traffic is intended to evaluate variants of the organization of these works, which considers the technical, cost, and safety aspects. Basic scientific methods of induction, deduction, abstraction, concretization, classification, and visualization were used. Qualitative and quantitative research, brainstorming, and consultations with opponents from practice were carried out. A key step in the proposed procedures is the use of procedures in the modeling of traffic processes, where simulation tools and operational analysis tools can be used. It is about determining the optimal algorithm or simulation model to facilitate the solution of procedures in cases of partial or complete blocking of the designated route. The essence is in the use of a mathematical apparatus, which has the task of analyzing the impact of interruptions, identifying affected train paths, quickly rescheduling train paths in real time after traffic interruptions, optimizing costs, and minimizing delays by adjusting timetables or routes. The models optimize the planning of train paths in the timetable, including heuristic algorithms, data analysis, and simulations, to minimize the impact of restrictions on rail traffic and ensure the safety and quality of railway services.

There are many tools, methods, and mathematical models that can be used to optimize railway operations during infrastructure interruption. The RailSys software 3.0 [53] is a planning system for railway transport, but at the same time, it provides a very flexible use of integration and functionality to consider all requirements and needs. It is suitable for infrastructure management, capacity management, timetable simulation, operational simulation, and others. Models such as [54,55] also serve to optimize and propose the train traffic diagram during emergency situations, such as infrastructure interruption. The authors of [54] describe models for the optimization of railway timetables and their applicability in practice. The authors of [55] describe a heuristic approach to the integration of train timetable planning, platform planning, and maintenance planning of the railway network. The authors of [56] describe a model for determining the price for the allocation of railway infrastructure capacity in the Czech Republic. There are still many more models for railway timetable optimization and planning [57–62].

We can also think of a railway network as a directed graph with nodes and edges with different weights, taking the nodes as branching railway stations and the edges as inter-station sections. Such a network can include simple requests but also inputs from advanced logistics systems. Connected transport operations are entering the system, increasing the

complexity of transport networks. More inputs equal a more demanding possibility of predicting the occurrence of a fault [63]. This is addressed as a simulation model exploring logistics networks. The output of the model is the determination of faults in the given network and the recommendation of their solution. The model works with three basic scenarios and allows for changes in requirements: first, if there is an increased demand during a certain period for a certain area; second, if there is an increased network-wide demand during a period; and third, if there is a permanent change in demand to find a new route with sufficient transport capacity. The creation of timetables is a complex process based on the experience and expertise of their creators and, at present, the great support of software tools. Creators face challenges to ensure that the constructed timetables are sufficiently robust (durable) and the accuracy of operation is maintained despite high demand and minimal capacity consumption.

The methodological procedure is verified using the example of the infrastructure of the Czech infrastructure manager. Railway infrastructure interruption is governed by internal regulations [35–41,64]. The methodology is established based on four types of proposed methods related to the railway infrastructure interruption, the determination of technological procedures for the interruption, and safety during the railway infrastructure interruption (Figure 13) as a tool for better management of the railway infrastructure interruption on the railway network. Subsequently, the individual procedures in the methodology are characterized.

Quantitative research was focused on measuring and testing data from real railway infrastructure interruptions on the solved railway lines and transport performance in railway transport. Tests of causal relationships between variables were performed, predictions were made, and results were generalized to passengers, operators, and the infrastructure manager. The literature review includes academic literature searches (peer-reviewed journal articles only) conducted on the Scopus and Google Scholar platforms, as well as scholarly article searches. The keyword combinations used to search the academic and grey literature were as follows:

(A) "railway interruption" OR "interruption" OR "railway operation" OR "response to interruption";
(B) "maintenance" OR "maintenance of railway infrastructure" OR "repair of railway infrastructure" OR "renewal" OR "strategy" OR "strategies";
(C) ("modernization" OR "modernization of railway infrastructure" OR "response to modernization";
(D) "planned interruption of railway traffic" OR "interruption of railway traffic strategy" OR "unplanned interruption of railway traffic";
(E) "railway infrastructure" OR "railway transport route" OR "railway line";
(F) "rail transport" OR "railway transport" OR "mobility";
(G) "Impact" OR "consequences" OR "conduct" OR "travel" OR "perception" OR "traffic".

The issue of railway infrastructure interruption is at a certain level in the academic literature, but the methodology and management of railway traffic during the infrastructure interruption are only a small scope of published research.

An important research procedure was the discussion of practitioners, where the impacts of planned and unplanned railway infrastructure interruptions were determined. The main brainstorming format was panel discussions with these experts, where it was determined what needed to be solved from the perspective of operators, infrastructure managers, and also passengers. Quantitative research was carried out using brainstorming methods, the aim of which was to propose methods for the management methodology during the railway infrastructure interruption. As part of the qualitative research step, we investigated the opinions, behaviors, and experiences of railway transport experts. They collected and analyzed data and findings in research papers. In this contribution, the results and findings for the railway lines České Budějovice–České Velenice and České Budějovice–Horní Dvořiště are used. A mixed-methods research methodology is used to confirm the findings and explain any results obtained to verify that the results observed using both

methods are complementary. Based on the mentioned methods, the authors proposed a universal methodology for the railway operation management system during the planned infrastructure interruption. There have been several useful scientific methods that are general and universal but are particularly applicable to a given issue. The methodology consists of several steps, which are shown in Figure 10.

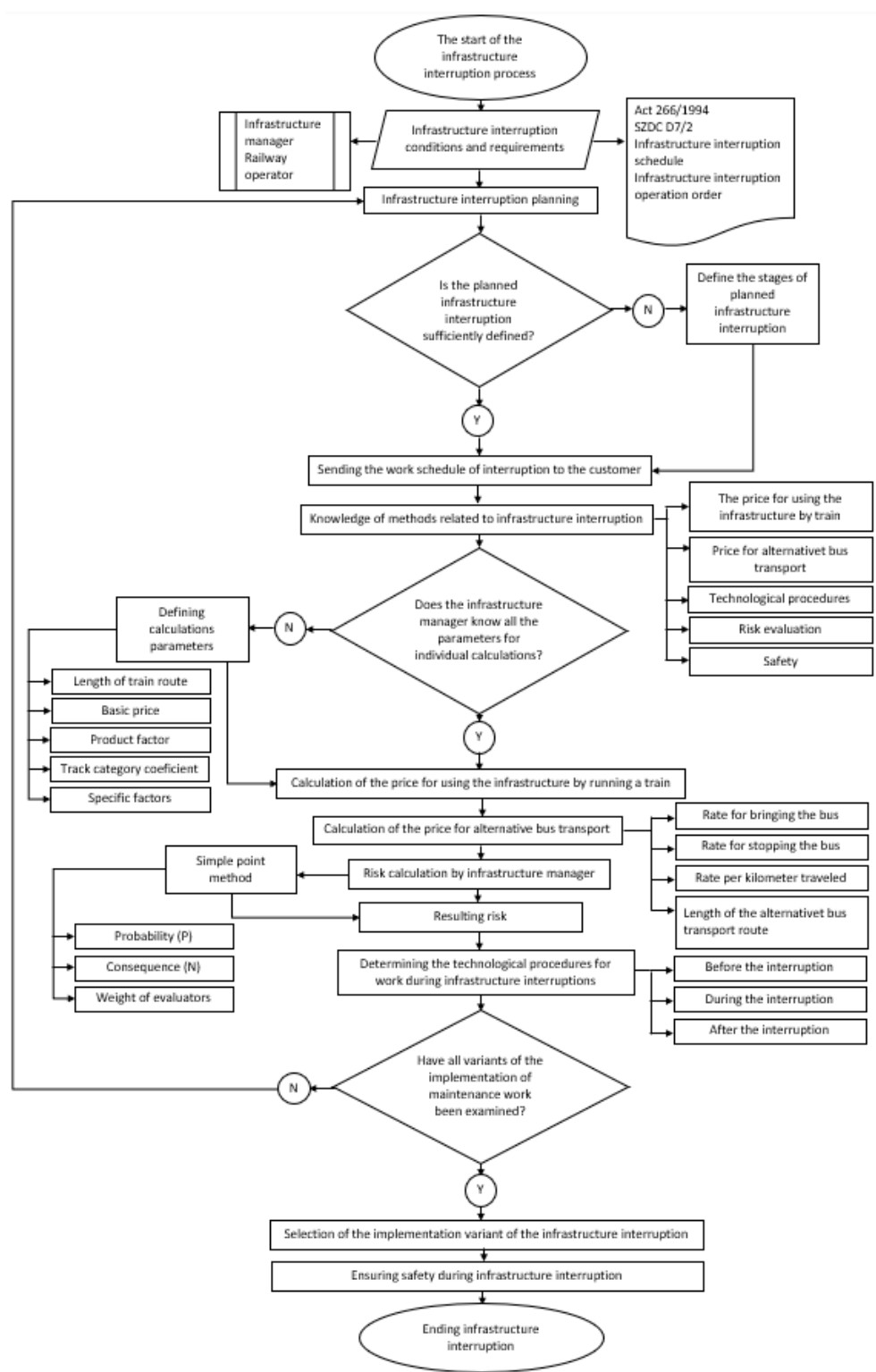

**Figure 13.** Work management methodology during the railway infrastructure interruption.

Defining the conditions and requirements of the planned interruption of traffic on the infrastructure. This concerns both the infrastructure manager and the railway undertakings (operators) and end customers (passengers, forwarders). The railway operator in passenger transport will ensure that passengers are informed about alternative bus transport, train queues, and the locations and marking of stations intended for boarding or disembarking passengers at the railway stations affected by the interruption of traffic. The railway operator in freight transport informs their customers about the restriction of loading and unloading at the affected transport points and the interruption of railway tracks.

Define the individual stages of the planned infrastructure interruption. For each stage of the planned infrastructure interruption, it is necessary to determine the range of days and the time range when the planned interruption will be implemented, define the specific sections that the planned interruption will affect within each stage, and determine the distances for rail transport and substitute bus transport for the subsequent calculation of prices within the methodology.

Submitting the work schedule during the planned interruption. The schedule of the planned interruption of the railway traffic of temporarily limited capacity consists of four levels of impacts on the time course of planning (low impact, medium impact, high impact, and significant impact). It is a system of coordination and consultation over the course of planning.

Knowledge of methods related to planned infrastructure interruption. These methods include the price for the use of the infrastructure, the price for alternative bus transport, the assessment of risks by the infrastructure manager, the determination of technological work procedures, and safety. In freight transport, it is necessary to determine diversion routes or stations where trains will wait for the interruption to end.

The calculation of the price for the use of the infrastructure. This is determined based on the established procedures of each railway infrastructure manager, which, in principle, differ very little. In terms of the Czech infrastructure manager, it is determined according to Formula (1), considering the track category, product factor, and specific factors.

The calculation of the price for alternative bus transport. The amount of this price is determined by RU based on the product of the relevant unit prices, the transport services performed, and the necessary waiting periods according to Formula (2). In case of using alternative bus transport, the railway operator saves the price for the use of railway infrastructure.

An evaluation of possible risks by the infrastructure manager and the calculation of risk using the simple point method—three components (probability, consequence, and evaluator weights) are evaluated according to Formula (3).

Determination of technological work procedures during the planned infrastructure interruption. This is a technological procedure of work that is divided into three categories and must be firmly established and observed regarding the timetable of the interruption of traffic. The idea is to ensure that the work on the track is done safely and in the shortest possible time.

Selection of a variant for the implementation of infrastructure interruption. In this step, based on the several variants presented for the implementation of the planned infrastructure interruption, the optimal variant will be selected. The decisive factor is the global view of the optimization criterion, which is the minimization of costs caused by works during the infrastructure interruption as the total costs of all actors, i.e., the manager of the railway infrastructure and the affected operators.

Ensuring safety during the implementation of the planned infrastructure interruption. All persons who participate in the implementation of the works will, in the process, implement the movement of railway vehicles and mechanisms along the track operated by IM. Also, all employees who participate in the operation of the railway and railway transport in connection with the implementation of maintenance. Safety must also be ensured for passengers (all construction measures necessary for the safe exit and boarding of passengers).

### 4.1. Price for Using the Infrastructure by Train

To determine the price for the use of railway infrastructure, each IM has established procedures that are very similar in principle. In terms of the Czech infrastructure manager, a calculation formula is used [65,66]:

$$C_{train} = L \times Z \times K \times P_x \times S_1 \times S_2 \tag{1}$$

where

$C_{train}$ denotes the price for using the infrastructure by running a train.
L denotes the length of the train route.
Z denotes the basic price.
K denotes the track category coefficient (Table 4).
$P_x$ denotes the product factor (Table 5).
$S_1$ and $S_2$ denote specific factors (Tables 6 and 7).

**Table 6.** Coefficient of line category K (authors, according to [64]).

| Line Category | Value K |
|:---:|:---:|
| 1 | 1.15 |
| 2 | 1.12 |
| 3 | 1.00 |
| 4 | 0.88 |
| 5 | 0.71 |

**Table 7.** Product factor P (authors, according to [64]).

| Product Factor | Value P |
|---|:---:|
| $P_1$—passenger transport | 1.00 |
| $P_2$—unspecified freight transport | 1.00 |
| $P_3$—freight transport within the collection and delivery system of individual wagon shipments | 0.30 |
| $P_4$—combined freight transport | 0.65 |
| $P_5$—freight transport—non-standard trains | 2.00 |

The categorization of individual lines depends on the result of the evaluation of the technical condition of the line, the type of security equipment, and the inclusion of the line in the TEN-T network. The line category coefficients are shown in Table 6.

The product factor is a factor that considers market segmentation into services with different price levels. Individual types of product factors are listed in Table 7.

The specific factor $S_1$ indicates the level of track wear depending on the total weight of the train. These factors are listed in Table 8.

**Table 8.** Rate of track wear depending on the total gross weight of the train (authors, according to [64]).

| Weight Range Brutto (t) | Value $S_1$ | Weight Range Brutto (t) | Value $S_2$ |
|---|:---:|---|:---:|
| to 49 | 0.42 | 1000–1199 | 2.77 |
| 50–99 | 0.49 | 1200–1399 | 3.36 |
| 100–199 | 0.59 | 1400–1599 | 3.88 |
| 200–299 | 0.76 | 1600–1799 | 4.36 |
| 300–399 | 0.94 | 1800–1999 | 4.89 |
| 400–499 | 1.14 | 2000–2199 | 5.37 |
| 500–599 | 1.34 | 2200–2399 | 5.92 |
| 600–699 | 1.50 | 2400–2599 | 6.39 |
| 700–799 | 1.76 | 2600–2799 | 6.88 |
| 800–899 | 2.03 | 2800–2999 | 7.30 |
| 900–999 | 2.31 | over 3000 | 8.35 |

The specific factor $S_2$ depends on the equipment of the active locomotive with the train safety system ETCS Level 2 or higher security equipment. The values for the specific factor $S_2$ are shown in Table 9.

**Table 9.** Specific factor $S_2$ (authors, according to [64]).

| Equipment of the Driving Vehicle ECTS Level 2 and Above | Value $S_2$ |
|---|---|
| Non-equipped drive vehicle | 1.00 |
| Equipped drive vehicle | 0.95 |

*4.2. Price for Alternative Bus Transport for Canceled Trains*

The total price is formed by the product of the respective unit prices, the transport services performed, and the necessary waiting times. A calculation formula is used to determine the price for the use of the track by the train [67]:

$$C_{bus} = C_p + C_o + L \times C_{km} \tag{2}$$

where

$C_{bus}$ denotes the price for alternative bus transport.
$C_p$ denotes the rate for bringing the bus.
$C_o$ denotes the rate for stopping the bus.
$C_{km}$ denotes the rate per kilometer traveled by the bus.
L denotes the length of the alternative bus transport route.

*4.3. Simple Point Method for Risk Calculation*

The simple point method for risk calculation of IM consists of the evaluation of three components: probability, consequence, and weight of evaluators. The probability estimate consists of the considered danger that may occur, and the probability estimate is determined by numbers 1–15, where the degree of danger is simply included. The probability (P) is shown in Table 10.

**Table 10.** Probabilities of danger occurrence (authors, according to [68]).

| Degree | Frequency of Occurrence | Severity of Consequences |
|---|---|---|
| 1 | Unlikely | A very small threat |
| 2 | Likely | Small threat |
| 3 | Very likely | Frequent threats |
| 4 | Highly probable | Continuous threat |
| 5 | Permanent | Critical threat |

Like probability, the category of consequences is also determined. For each situation, the most serious possible damage will be determined using five degrees, which indicate the severity of the possible injury. The severity of the consequences (N) is shown in Table 11.

The product of the relevant values of severity and probability yields the final value of the risk of health damage (R), which is a risk.

Evaluation:

1. Negligible impact on the degree of danger and threat;
2. Little impact on the level of danger and threat;
3. Greater, non-negligible impact on the degree of danger and threat;
4. A large and significant impact on the degree of danger and threat;
5. More significant and adverse impacts on the severity and consequences of threats and hazards.

**Table 11.** Degrees of consequences (authors, according to [68]).

| Degree | Consequence | Example |
|---|---|---|
| 1 | Insignificant | Injury without incapacity for work, negligible system failure, incurred damage up to EUR 395.77, outage less than 1 day. |
| 2 | Small impact | Injury with incapacity for work, without permanent consequences, amount of damage (EUR 395.77–EUR 19,788.66), production outage of 1 day–2 weeks. |
| 3 | Bigger, not insignificant | Injury with permanent consequences (serious injury) that require long-term treatment, occupational disease, significant damage to the system, loss in production, large financial losses, damage ranges from EUR 19,788.66 to EUR 98,943.29, production outage of 2 weeks to 1 month. |
| 4 | Large and important | Large and significant influence on the degree of danger and threat (severe occupational accident). Extensive damage to the system, loss of production, large financial losses, damage is in the range of EUR 98,943.29–EUR 197,886.57, production outage lasts 1 month–4 months. |
| 5 | Catastrophic | Fatal injury, complete destruction of the system, irreplaceable losses, significant damage, the amount of damage is more than EUR 197,886.57, production outage lasts longer than 4 months. |

Risk calculation using a simple point method

$$R = N \times P \times H \tag{3}$$

where

R = risk.
N = consequence of the threat.
P = probability.
H = evaluation.

Based on the above calculation, we obtain the resulting risk. The resulting risk is divided into five categories and is shown in Table 12.

**Table 12.** Result matrix for risk assessment (authors, according to [68]).

| Category | Level of Risk | Risk Assessment |
|---|---|---|
| I. | Insignificant R is less than 5 | No action is required. |
| II. | Acceptable R is in the range of 6–10 | No additional management is required. Attention should be paid to improving solutions that would be more effective in terms of the resources spent and do not carry the burden of additional costs. Monitoring is required for procedural compliance. |
| III. | A slight risk R is in the range of 11–50 | It is necessary to minimize the risks that have arisen, and the costs of prevention need to be considered and defined. Risk minimization measures are to be implemented within the specified time. If a moderate risk is associated with very harmful consequences, further estimation and a more precise determination of the probability may be needed as a basis for determining the need to improve the management system. |
| IV. | Unwanted risk R is in range of 51–100 | Work should not be started until the risk has been reduced. Considerable resources can be allocated to risk reduction. If the risk-related work has already started, the necessary steps must be taken. |
| V. | Unacceptable risk R is greater than 101–125 | Work must not be started or continued until the risk has been reduced. If it is not possible to reduce the risk even by using unlimited resources, the work must be prohibited. |

*4.4. Technological Procedures during the Railway Infrastructure Interruption*

It is a technological procedure of work that is divided into three categories and must be firmly set and followed due to the time schedule of traffic interruptions. The point is that the work on the track is safe and carried out in the shortest possible time.

Determination of the technological procedure before the railway infrastructure interruption

- Measurements of vehicle runs are made on the tracks at regular intervals. The measurements' outcomes form the foundation for commissioning rail equipment maintenance and repairs, such as kilometer position specification;
- We carry out preparatory work before commencing the maintenance. It goes, for example, to the timely arrival of mechanization and delivery of material and necessary portable signs to a railway station to which the maintenance works are concerned. All employees participating in these projects must be informed about their technological progress. And that is in accordance with all applicable regulations. At the same time, they must be instructed about work safety and confirm their acknowledgment by signing;
- If mechanization is to be worked on during the maintenance work, which will have to be shut down at a certain station during this work, the contractor must request the allocation of tracks for these mechanisms through the operational application (in the case of the Czech infrastructure manager, it is the KAZAS application) well in advance.

Determination of the technological procedure during the railway infrastructure interruption

- Notification of the responsible representative of the maintenance work order to the transport employee that the conditions for starting the maintenance work are met, the preparatory work is completed, and the maintenance can be started;
- Beginning of maintenance—according to the interrupted type of equipment on the track, safety equipment, or voltage equipment;
- In the case of electrified lines and voltage interruption of traffic, switching off the current of the traction line;
- Possible switching off the crossing security device;
- Taking over the station, determining and marking the section with interrupted operation with portable signs;
- The arrival of mechanization in the workplace;
- Performing preparatory work for a specific activity stipulated by the relevant IM regulation (SŽ S3/1), the contractor is responsible for the timely and high-quality performance of preparatory work unless otherwise stipulated by the contract;
- Performing work according to the technological procedures specified in the IM regulation (SŽ S3/1);
- Completion of work;
- The return of mechanization from the workplace;
- Clearance of portable signs;
- Turning on and testing the correct function of the crossing security device;
- Switching on the current and testing the traction line;
- The responsible representative of the maintenance customer shall keep a record of the track availability and operability of affected track parts;
- Ending of maintenance.

Determination of the technological procedure after the end of the railway infrastructure interruption

- After the maintenance is completed, the traffic employee must check all indications of the security device and remove the temporary signs.

*4.5. Selection of the Variant of Implementation of Railway Infrastructure Interruption*

In this step, based on the several variants presented for the implementation of the planned infrastructure interruption, the optimal variant will be selected. The decisive factor is the global view of the optimization criterion, which is the minimization of costs caused by works during infrastructure interruption as the total costs of all actors, i.e., the manager of the railway infrastructure and the affected operators. On the IM side, these are additional costs due to the implementation of these works, including the costs of modifying the security equipment. On the part of the operators, these are additional costs for running trains on diversion routes and waiting for the end of planned interruptions or moods for alternative bus transport minus the costs of unrealized train routes and compensation from IM, according to Formula (4). The variant that shows the lowest costs will be implemented.

$$C_{IM} \mp C_{RU} = min \tag{4}$$

where

$\Delta C_{IM}$ denotes the additional costs of IM.
$\Delta C_{RU}$ denotes the additional costs of RU.

The infrastructure manager's additional costs are, therefore, all the costs that need to be incurred in connection with the interruption compared to the null option (no interruption). These are the costs of securing the work according to the chosen variant of technological procedures regarding risk. The carrier's additional costs are made up of the costs of alternative bus transport minus the costs of unrealized train paths. In the case of a freight operator, these are additional costs of diversion routes or incurred costs of trains waiting to end the interruption (energy consumption, train crews, etc.).

The detailed range of types of costs that need to be included in the calculation is presented in Table 13.

**Table 13.** Additional structure costs for calculation.

| | | |
|---|---|---|
| Additional costs of IM $\Delta C_{IM}$ | + | Total costs for securing the selected variant of exclusion<br>Discounts on the price of using the infrastructure on diversion routes for passenger trains<br>Discounts on the price of using the infrastructure on diversion routes for freight transport<br>Costs related to alternative bus transport (informing passengers in stations, information systems) |
| | − | Additional costs for extending the infrastructure interruption (labor costs)<br>Risk of injuries and accidents (costs for accidents during working hours) |
| Additional costs of RU $\Delta C_{RU}$ | + | Costs of alternative bus transport<br>The cost of the diversion routing<br>Costs of waiting for trains to end the interruption (train staff, energy, etc.)<br>Reimbursed fare costs due to interruption (passenger transport)<br>Costs of unrealized transports due to interruption (freight transport) |
| | − | The price for using the infrastructure for unrealized train paths |

*4.6. Safety during the Implementation of Railway Infrastructure Interruption*

All persons who participate in the implementation of the works and will, in the process, implement the movement of railway vehicles and mechanisms on the track operated by IM must have a contract with IM on the operation of railway transport. All employees who participate in the operation of the railway and railway transport in connection with the implementation of maintenance are required to have valid professional and medical

qualifications. Even in the maintenance planning phase, the contractor must provide the customer with a detailed work schedule, including a list of mechanizations that, during the performance of the work, restrict the operation of rail traffic on the adjacent track (slow runs, prohibition of trains with exceeded loading width, interruption of the adjacent track, etc.), including the deadline for when operation will be restricted.

Safety must also be ensured for passengers. All construction measures necessary for the safe exit and boarding of passengers (preserving the prescribed length of the platform, building access roads, maintaining crossings over the interrupted track, etc.), which are determined by valid documents and regulations, must be implemented in a timely manner and completely.

## 5. Results

As part of the application part of this research, the railway infrastructure interruption on the Czech infrastructure manager network, specifically in the region of České Budějovice, on the Horní Dvořiště–České Budějovice (706) and České Velenice–České Budějovice (705) lines are addressed. These lines are single-track, electrified with an AC traction system of 25 kW, and are included in the TEN-T system. Table 14 shows an overview of the individual stages of infrastructure interruption on line 705 and the number of affected trains operating on this line.

**Table 14.** Overview of the stages of infrastructure interruption on line 705 and the number of affected trains.

| 705 | | | Stage A | Stage B | Stage C | Stage D | Stage E |
|---|---|---|---|---|---|---|---|
| Infrastructure interruption duration (time) | | | 7:30 A.M.–3:35 P.M. | 7:20 A.M.–3:30 P.M. | 7:10 A.M.–3:20 P.M. | 7:15 A.M.–2:10 P.M. | 7:20 A.M.–2:00 P.M. |
| Duration of stages (days) | | | 2 | 2 | 2 | 2 | 2 |
| Operated regional trains | workday | even direction | 14 | 14 | 14 | 14 | 14 |
| | | odd direction | 13 | 13 | 13 | 13 | 13 |
| | weekend | even direction | 11 | 11 | 11 | 11 | 11 |
| | | odd direction | 10 | 10 | 10 | 10 | 10 |
| Replaced trains | workday | even direction | 4 | 4 | 4 | 4 | 3 |
| | | odd direction | 4 | 4 | 4 | 4 | 3 |
| Freight trains | | even direction | 1 | 1 | 1 | 1 | 1 |
| | | odd direction | 1 | 1 | 1 | 1 | 1 |

Table 15 shows an overview of the individual stages of infrastructure interruption on line 706 and the number of affected trains.

**Table 15.** Overview of the stages of infrastructure interruption on line 706 and the number of affected trains.

| 706 | | | Stage A | Stage B | Stage C+F | Stage D | Stage E | Stage F |
|---|---|---|---|---|---|---|---|---|
| Infrastructure interruption duration (time) | | | 9:10 A.M.–4:45 P.M. | 9:00 A.M.–4:25 P.M. | 8:45 A.M.–4:15 P.M. | 8:50 A.M.–4:10 P.M. | 8:55 A.M.–4:00 P.M. | 8:45 A.M.–4:15 P.M. |
| Duration of stages (days) | | | 2 | 3 | 2 | 2 | 1 | 1 |
| Operated regional trains | workday | even direction | 10 | 10 | 10 | 10 | 10 | 10 |
| | | odd direction | 11 | 11 | 11 | 11 | 11 | 11 |
| | weekend | even direction | 8 | 8 | 8 | 8 | 8 | 8 |
| | | odd direction | 9 | 9 | 9 | 9 | 9 | 9 |
| Operated fast trains | workday | even direction | 4 | 4 | 4 | 4 | 4 | 4 |
| | | odd direction | 4 | 4 | 4 | 4 | 4 | 4 |
| | weekend | even direction | 4 | 4 | 4 | 4 | 4 | 4 |
| | | odd direction | 4 | 4 | 4 | 4 | 4 | 4 |
| Replaced trains | weekend | even direction | 4 | 4 | 4 | 4 | 4 | 4 |
| | | odd direction | 4 | 4 | 4 | 4 | 4 | 4 |
| Freight trains | | even direction | 1 | 1 | 1 | 1 | 1 | 1 |
| | | odd direction | 2 | 2 | 2 | 2 | 2 | 2 |

Figure 14 shows the schedule of planned infrastructure interruption on lines 705 and 706.

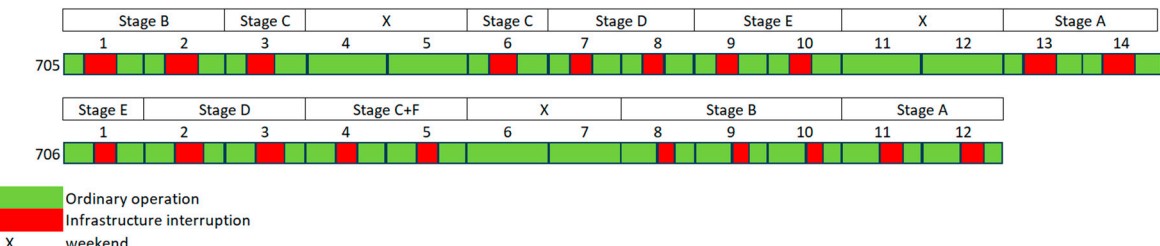

**Figure 14.** Time schedule of planned infrastructure interruption.

The individual stages are arranged according to the day of their implementation and not according to the designation. This designation is only formal. Maintenance activities were carried out gradually and were assigned to individual stages according to the importance of its implementation. The green color shows the normal operation of railway transport without restrictions, and the red color shows the duration of the infrastructure interruption (these times are shown in Tables 14 and 15). After the end of the maintenance activity, the operation was resumed until the next day, when the operation was carried out again at the specific time of the planned maintenance. On line 705, the planned infrastructure interruption lasted 14 days (including two weekends when the operation was in normal mode—they are marked with an X). On line 706, the planned interruption lasted 12 days (including one weekend—operation in normal mode). The total time of railway infrastructure interruption during the entire period was 38 h for line 705 and 44 h and 25 min for line 706.

### 5.1. Operator Costs during Infrastructure Interruptions

The necessary parameters of abandoned passenger trains on railway line 705 and the parameters of canceled regional trains (Reg) and fast trains (F) on railway line 706 were obtained from the operational application train track position. Table 16 shows the prices for the use of the track by the train for all stages of infrastructure interruption, which are calculated according to Formula (1).

**Table 16.** Price for using the track by train.

| Stage | | A | | B, C | | D, E | | Infrastructure Interruption Operation Order | |
|---|---|---|---|---|---|---|---|---|---|
| | Train category | 705 | 706 | 705 | 706 | 705 | 706 | 705 | 706 |
| L [kilometer] | Reg | | 7.3 | | 35.8 | | 8.1 | | |
| | F | 13 | - | 16.4 | 16.7 | 19.9 | 8.1 | | |
| | Ex | | - | | 24 | | 56.8 | | |
| Z [EUR] | | 0.85 | 0.85 | 0.85 | 0.85 | 0.85 | 0.85 | | |
| K | | 1 | 1 | 1 | 1 | 1 | 1 | | |
| P1 | | 1 | 1 | 1 | 1 | 1 | 1 | | |
| $S_1$ | Reg | 0.76 | 0.59 | 0.76 | 0.59 | 0.76 | 0.59 | | |
| | F | - | 0.76 | - | 0.76 | - | 0.76 | | |
| | Ex | - | 0.94 | - | 0.94 | - | 0.94 | | |
| $S_2$ | Reg | 1 | 1 | 1 | 1 | 1 | 1 | | |
| | F | - | 1 | - | 1 | - | 1 | | |
| | Ex | - | 1 | - | 1 | - | 1 | | |
| C/train [EUR] | Reg | 8.41 | 3.66 | 10.61 | 17.97 | 12.87 | 4.07 | | |
| | F | - | 4.72 | - | 10.80 | - | 5.24 | | |
| | Ex | - | 5.84 | - | 18.13 | - | 45.43 | | |
| C/stage [EUR] | | 134.51 | 791.03 | 169.69 | 259.47 | 154.43 | 288.25 | 458.63 | 579.03 |

For trains that are replaced by alternative bus transport due to maintenance, the operator would pay the sum of EUR 458.63 for railway line 705 and EUR 579.03 for railway line 706 when using the railway transport route. However, the operator thus incurred costs associated with the provision of alternative bus transport.

### 5.2. Price for Alternative Transport for Canceled Trains

The calculation of the price, according to Formula (2), for the operator of alternative bus transport for one bus is made up of the price for bringing in and stopping the bus; the product of the relevant unit price, which is EUR 2.77/train kilometer; and the performed transport services. The assumption is that each train is replaced by two buses. Table 17 shows the price for alternative bus transport and the increase compared to train travel.

**Table 17.** Price for alternative bus transport for canceled trains on railway line 705.

| Stage | L [km] | C/BUS [EUR] | C/Stage [EUR] | Increase [EUR] |
|---|---|---|---|---|
| A | 14.7 | 46.27 | 740.25 | |
| B, C | 28.1 | 83.39 | 1334.23 | |
| D, E | 24.9 | 74.52 | 1192.39 | |
| Infrastructure interruption operation order | | | 3266.87 | 2808.23 |

The railway operator will pay EUR 3266.87 for alternative bus transport, which is an increase of EUR 2808.23. Figure 10 shows a comparison of the prices for the transportation route traveled by trains and buses, as well as alternative bus transport, in individual stages. The resulting price is calculated according to the calculation in Formula (2). Table 18 shows the price for alternative bus transport and the increase compared to train travel.

**Table 18.** Price for canceled trains on railway line 706.

| Stage | Rail Infrastructure | Alternative Bus Transport | Increase [EUR] |
|---|---|---|---|
| A | 31.31 | 806.74 | |
| B | 172.20 | 3457.47 | |
| C | 87.27 | 2637.99 | |
| D | 156.10 | 2770.41 | |
| E | 132.15 | 2537.70 | |
| Infrastructure interruption operation order [EUR] | 579.03 | 12,210.27 | 11,631.28 |

The price for the railway journey covered by trains would be EUR 579.03. Since many buses were used, the costs for alternative bus transport reached EUR 12,210.27. The operator's costs were increased by EUR 11,631.28. The comparison of the price for the traveled transport path of railway lines 705 and 706 is shown in Figure 15.

It is obvious that the railway transport operator will pay several times more than for normal operation when trains are running in the given section. The increase in these costs is EUR 2808.23 during the infrastructure interruption. However, this price is not final. The railway operator pays the railway transport operator compensation for this interruption. From a logistical and cost point of view, it is more advantageous for operators to have infrastructure interruptions that do not result in a complete stoppage of operations or the use of alternative bus transport.

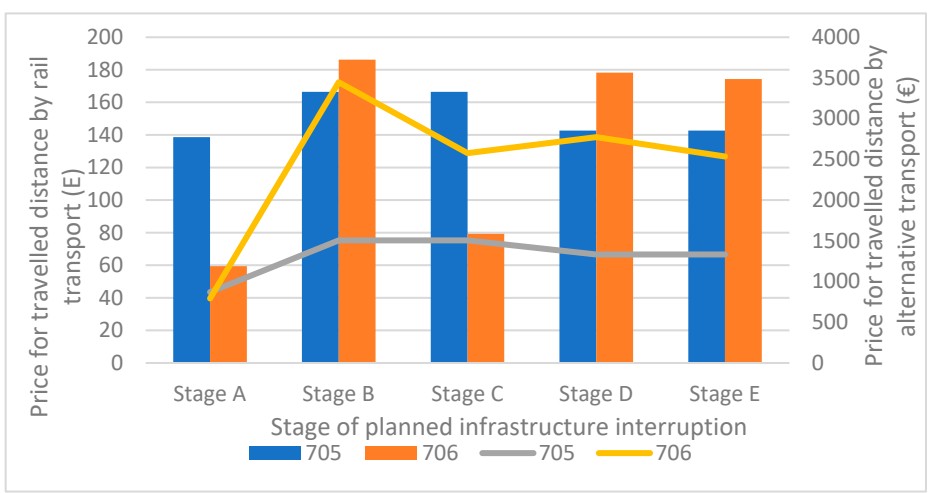

**Figure 15.** Price for the traveled transport path of railway lines 705 and 706.

### 5.3. Simple Point Method

This method is applied to all activities and causes of risk. As an example, the situation of the movement of workers in the rail yard is given. During the movement of workers on the track, the worker may be hit by railway vehicles, slip, fall, or be injured by falling objects.

According to the interviewing experts, the probability that this risk may occur is set to 2 because it is a probable phenomenon that can occur. The consequence of this risk is assessed as level 3 because an injury with permanent consequences (a more severe injury) may occur, which requires long-term treatment. Extensive damage may even occur. The economic evaluation in the amount of EUR 98,943.29– EUR 197,886.57 may include the treatment of the employee and the damage to property that will be caused by this risk. The weight for the evaluators was set to 4—a large and significant impact on the degree of danger and threat. Assessors set this value because these injuries happen. The resulting risk is calculated using a simple point method according to Formula (3). The resulting risk will, therefore, be calculated as follows:

$$R = 2 \times 3 \times 4 = 24$$

The resulting risk rate is 24, which belongs to category 3—moderate risk. As a measure to reduce this risk, the following will be established:

- Use prescribed and assigned protective equipment in proper condition (reflective protective equipment);
- Enter the track concentratedly and only for the performance of work;
- Move with increased caution, always anticipate the movement of rolling stock, and look around in all directions;
- Use designated paths;
- Cross the tracks only in places reserved for this (crossings, footbridges, underpasses);
- Increase attention when passing railway vehicles; step away from moving railway vehicles;
- Use the assigned lights in reduced visibility;
- Adjust the walking speed about the possibility of the terrain, considering the current state of the weather and the possibility of an injury;
- Prohibition of the movement of persons in places marked with safety signs "Prohibition of entry and movement of persons";
- Securing dangerous openings and depressions;
- Ensuring sufficient free space for the movement of people along the tracks;
- Professional and medical qualifications of employees;

- Qualified staff;
- Ban on the consumption of alcohol and narcotic substances.

### 5.4. Costs of the Railway Operator during the Railway Infrastructure Interruption and Technological Procedures of Works

If the operator must provide alternative bus transport due to maintenance on the transport route, the railway operator pays compensation, which is one of the cost items. The amount of compensation is non-public information. The costs of the railway operator include, among other things, the wages of employees.

The usual working hours of the organizational units ensuring maintenance are 6 A.M.–2 P.M. In the case of resolved maintenance, which always took place outside working hours, the employees are entitled to an hourly wage increased by additional payments according to the collective agreement. Namely, a 25% surcharge for overtime and EUR 0.28 for each hour. Wages up to 2 P.M. cannot be included in the costs because, at this time, regular IM employees are paid even if there is no infrastructure interruption. Table 19 shows the labor costs incurred within the individual stages of infrastructure interruption.

**Table 19.** Additional labor costs for IM employees within the individual stages of work during infrastructure interruptions.

| Tariff Level/ Number | Rail Line | Stage A [EUR] | Stage B [EUR] | Stage C, F [EUR] | Stage D [EUR] | Stage E [EUR] | Infrastructure Interruption Operation Order |
|---|---|---|---|---|---|---|---|
| 5/2 | 705 | 52.08 | 48.98 | 44.74 | 14.46 | 10.47 | |
| | 706 | 82.36 | 109.17 | 68.79 | 66.55 | 31.28 | |
| 6/4 | 705 | 113.19 | 106.62 | 97.38 | 31.54 | 22.84 | |
| | 706 | 179.03 | 237.49 | 149.63 | 144.73 | 68.02 | |
| 9/3 | 705 | 105.61 | 99.86 | 91.16 | 29.66 | 21.48 | |
| | 706 | 167.11 | 222.09 | 139.88 | 135.28 | 63.55 | |
| 10/2 | 705 | 75.32 | 71.28 | 65.07 | 21.20 | 15.35 | |
| | 706 | 3011.50 | 158.48 | 99.80 | 96.51 | 45.33 | |
| 12/1 | 705 | 119.19 | 40.37 | 36.85 | 12.03 | 8.71 | |
| | 706 | 67.40 | 89.70 | 56.48 | 54.61 | 51.30 | |
| Total [EUR] | 705 | 388.79 | 367.12 | 335.20 | 108.89 | 78.85 | 1278.85 |
| | 706 | 615.10 | 816.92 | 514.58 | 497.68 | 259.47 | 2703.75 |

The amount of labor costs in case of interruption on railway line 706 is EUR 2703.75, and on railway line 705 it is EUR 1278.85.

### 5.5. Work Safety during the Infrastructure Interruption

Maintenance works are activities performed on or near the track. The work is governed by safety regulations, where the risks are determined. There were three occupational accidents during the interruption on railway line 705. One was classified as a serious work injury, and the other was classified as a minor work injury. The cause of the accident was an unforeseeable occupational risk. During the interruption on railway line 706, there was one occupational accident, which was evaluated as a serious occupational accident. A worker was injured by a work machine.

## 6. Discussion

Currently, maintenance activity is increasing, and certain risks are associated with it. Based on the analysis performed, deficiencies were identified, especially in the observance of occupational safety and related risks. During the maintenance of the treated track sections, there was an increase in the number of occupational accidents. In connection with this, costs have increased, but it also affects the course of maintenance activities.

In the application part, two track sections were selected. The infrastructure interruption took place on the České Velenice–České Budějovice (railway line 705) and on the Horní Dvořiště–České Budějovice (railway line 706) lines. On both track sections, repair and maintenance work was carried out on the top and bottom of the railway in the track sections. During both maintenance operations on these track sections, according to the definition in the individual stages, the operation on the railway transport route was completely stopped. To preserve the public service, the operator had to introduce measures in the form of alternative bus transport for passenger transport.

The economic point of view of the investigated maintenance activities showed the possibility of reducing costs. One possibility is to move maintenance activities to night hours. During normal operation, the operator paid EUR 458.63 for trains running on the České Velenice–České Budějovice line. As part of the maintenance, the operator must pay EUR 2808.23 for alternative bus transport. However, if the maintenance was shifted to night hours, only three trains would be affected by this maintenance. The operator would pay EUR 1225.08 for alternative bus transport. Savings would also occur in wage costs for employees of the infrastructure manager working on maintenance. The employer will pay additional fees in the amount of EUR 1278.85 to the employees for maintenance at the specified time. The time shift would also be suitable from an economic point of view for the maintenance of the 706 Horní Dvořiště–České Budějovice line. On this line, the operator will pay EUR 579.03 for the canceled train paths on the railway infrastructure. Since it is a line where fast trains also run, in case of stopped operation and use of alternative bus transport, the operator will pay EUR 12,210.31. With the time shift, there would also be a reduction in the number of trains that are replaced. The operator will pay EUR 369.14 for trains running on the railway infrastructure. The operator will pay EUR 8156.09 for alternative bus transport, replacing these trains. Considering the time frame of maintenance in individual stages and the work of employees beyond working hours, the wage costs are EUR 2703.74.

In terms of safety, there were a total of four occupational accidents in the analyzed maintenance activities. Two accidents were classified as minor occupational accidents; the other two accidents were classified as severe occupational accidents. Based on the analysis, it was found that the biggest problems are outdated work risks and non-compliance with established regulations. Subsequently, it is recommended that the date of training in safety and health protection at work be set at an interval of once a year. Currently, training takes place once every two years.

Infrastructure interruption during the night hours was not addressed and calculated in detail in this article. As part of the comparison of possible solutions for the introduction of night interruptions on the infrastructure, the operator would pay EUR 1225.08 on railway line 705 and EUR 8156.09 on railway line 706. The railway infrastructure manager would pay EUR 1143.74 on railway line 706 as part of night-time infrastructure interruptions on railway line 705. As part of the economic benefits, shifting the time of implementation of infrastructure interruptions to the night hours would lead to financial savings, both for the infrastructure manager and for the operators. Table 20 presents the costs for alternative bus transport for operators and wage costs for employees of the infrastructure manager for individual maintenance activities in the current time frame, in case of a possible time shift, and the amount of savings.

**Table 20.** Costs and savings for alternative bus transport and labor costs.

| Interrupted Railway Line | Operator—Costs for Alternative Bus Transport [EUR] | | | IM's Additional Labor Costs [EUR] | | |
|---|---|---|---|---|---|---|
| | Existing | Night Time | Savings | Existing | Night Time | Savings |
| 705 | 3266.87 | 1225.08 | 2041.79 | 1278.85 | 1143.74 | 135.11 |
| 706 | 12,210.31 | 8156.09 | 4054.22 | 2703.75 | 1143.74 | 1560.00 |



The operator's savings for maintenance on the České Velenice–České Budějovice line amount to EUR 2041.79, and on the Horní Dvořiště–České Budějovice line, they amount to EUR 4054.22. The infrastructure manager's savings for maintenance on the České Velenice–České Budějovice line amount to EUR 135.11, and on the Horní Dvořiště–České Budějovice line, they amount to EUR 1560.00. Operator costs for line 705 are EUR 326.67 per day, and for line 706, they are EUR 1221.03 per day. The infrastructure manager's costs for line 705 are EUR 127.88 per day for maintenance activities, and for line 706, they are EUR 270.37 per day.

Trouble-free operation on the railway requires regular maintenance of the infrastructure, while according to the findings, up to 25% of the length of the infrastructure is subject to some activities aimed at maintenance every year and, with it, planned interruption of traffic on the infrastructure. Maintenance interventions interfere with normal operation and limit the fulfillment of the established timetable. The proposed methodology and its individual procedures (calculations, technological procedures, and others) represent a suitable tool for the infrastructure manager to properly manage the interruption of railway traffic on the infrastructure. The application part of this research is proposed based on real infrastructure interruptions on the network of the Czech infrastructure manager. It follows from the proposed methodology that the economic point of view of the investigated maintenance activities showed the possibility of reducing costs, proposing the possibility of implementing infrastructure interruption during night hours and stricter compliance with safety at work to reduce the risks of accidents as a tool for reducing costs, which also affects the course of maintenance activities. Since the operation and maintenance of railway infrastructure require a long-term and sustainable strategy, it is important to set up infrastructure interruption management to ensure regular, reliable, and safe railway infrastructure for all operators.

The emphasis in the research was placed on the economic side of this issue and a global view of the railway market and the provision of the resulting service at the required quality level. Another key moment in the methodology is the precise planning of the travel schedule during infrastructure interruptions using modeling and simulation. There may be situations where such a timetable is not able to handle existing requests, e.g., due to the delay of the trains concerned and other circumstances. The situation created in this way must be treated as an unplanned interruption. These newly created interruptions cause deviations in timetables during infrastructure interruptions and cause delays at destination stations. The aim is for the timetable during the infrastructure interruption to be closer to the original timetable and to cover, as much as possible, all existing rail traffic requirements. A correct understanding of the technological aspects of the planning of operational activities has a decisive influence on the economic efficiency of the evaluated measures during infrastructure interruption.

Passenger transport also incurs increased costs for passengers in terms of increased waiting time, longer transport time, or discomfort. The presented methodology does not calculate the cost of lost passenger time. This approach would be possible if we were to calculate the societal (economic) costs of different variants of railway operation. In that methodology, returned tickets are counted as giving up the intended trip due to the interruption. Similarly, in freight transport, the costs of foregone transports due to road closures or the reduced transport capacity of the train are included (reduced standard weight of the set due to the deviation along a more inclined track) or additional costs for maintaining the transport capacity of the train (for example, the addition of a spur locomotive or an independent locomotive traction).

Compared to the research of [18], where the authors dealt with the proposal of an alternative timetable in case of interruption of the railway infrastructure due to maintenance activities, our paper does not deal with a specific proposal of the timetable in the case of infrastructure interruption but proposes a management system of infrastructure interruption from the point of view of the infrastructure manager and the operator and costs incurred for this infrastructure interruption. Our research should be understood

from the point of view of saving costs, ensuring safety during maintenance activities, and ensuring the transportation of passengers during the duration of the infrastructure interruption. Compared to the research of [25], where the authors dealt with the management of the interruption of railway operations, we can state that this research mainly focused on passengers and their transportation options. This research dealt with a strategy focused on travelers. Compared to this research, our paper mainly focused on the infrastructure managers and operators who are directly affected by the infrastructure interruption. In the case of further research, it would be appropriate to link our research together with the proposal of a strategy focused on passengers and their transport during infrastructure interruptions and link these two strategies into a suitable model.

The originality of this research comes in solving the problem of planned infrastructure interruption from the point of view of the infrastructure manager, as well as the costs and safety measures spent on employees who perform maintenance activities. This paper also deals with the cost activity of railway operators, which are directly affected by the infrastructure interruption by the fact that their performance is transformed into the performance of alternative bus transport. In the case of freight transport, the carrier's risk of shipment delays increased, as diversion routes were not used, and all freight trains (line 705—two freight trains; line 706—three freight trains) waited at stations until the end of the infrastructure interruption. Also, the originality of this research is focused on the safety of employees of infrastructure managers at work in the framework of reducing the risk of possible accidents and injuries.

## 7. Conclusions

Ensuring the operability, maintenance, and care of the transport infrastructure is the duty of the IM in terms of the EU transport policy. The fulfillment of this requirement is determined precisely by works on the transport infrastructure, which can be carried out in two ways. The first method is the restriction of operation, and the second method is the interruption of traffic on the infrastructure. In any case, it is very important that the maintenance activity is carried out in an optimized manner with a global view towards all actors in the railway market, minimizing the costs caused by this activity.

The goal of the research was, based on the analysis of the current state, to evaluate the works during the infrastructure interruption and to determine the most advantageous variant of the implementation of these measures. The evaluation was carried out in terms of the economic and safety impacts of maintenance activities. The proposal consists of the processing of methodological procedures that lead to the efficiency of work during the infrastructure interruption from a global perspective. The methodology can contribute to determining the amount of compensation to operators operating passenger transport for complications in the introduction of substitute bus transport. This paper deals with the activities during the infrastructure interruption of the Czech infrastructure manager, other activities, and the risks associated with them. In the application part, an analysis of maintenance activities on the lines České Velenice–České Budějovice and Horní Dvořiště–České Budějovice was carried out. As part of both maintenance activities, a calculation was made according to calculation formulas for the timing of maintenance activities, as well as subsequently if maintenance is shifted to night hours. It was found that the amounts paid by the operator for the railway route traveled and the amounts paid for alternative bus transport were subsequently compared. In the case of the infrastructure manager, the calculation was based on the wage costs of employees working on maintenance with the current time allocation and shift to evening and night hours. From an economic point of view, a time shift is clearly appropriate for both maintenances. And that is both from the point of view of the operator and from the point of view of the infrastructure manager. In terms of time, the shift would also increase the quality of transport services, as fewer trains would be affected by maintenance. In this way, inconvenience to passengers associated with alternative bus transport would be eliminated to a minimum.

**Author Contributions:** Conceptualization, Z.B. and J.G.; methodology, Z.B. and J.G; validation, J.G.; formal analysis, Z.B.; investigation, J.G.; resources, Z.B.; data curation, V.Z.; writing—original draft preparation, Z.B. and J.G.; writing—review and editing, Z.B.; visualization, V.Z.; supervision, J.G. and V.Z.; project administration, J.G.; funding acquisition, J.G. All authors have read and agreed to the published version of the manuscript.

**Funding:** This research received no external funding.

**Data Availability Statement:** This study did not report any specific data.

**Acknowledgments:** This paper is supported by the VEGA Agency through Project 1/0640/23, "Elements of quality in competitive public tendering in railway passenger transport", through the Faculty of Operations and Economics of Transport and Communication, University of Žilina.

**Conflicts of Interest:** The authors declare no conflicts of interest.

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
