# Peer review of "The Management of Railway Operations during the Planned Interruption of Railway Infrastructure"

_infrastructures, doi:10.3390/infrastructures9070119_

Round 1

Reviewer 1 Report

Comments and Suggestions for Authors

(1) In the abstract, it is suggested to change ‘track line or the operation of …’ to ‘railway infrastructure’, as the interruption is used to maintain the infrastructure instead of just the track.

(2) Please include the main method used in this paper in the abstract to highlight the scientific contribution of this paper. For the current abstract, it looks like a case study report.

(3) Again, the main theoretical method should be included in the keywords.

(4) Please distinguish the difference between the railway infrastructure and the track. Normally, there are two parts for the railway infrastructure, namely, the track and the overhead system [1-3]. It is recommended to include the following references in the introduction to briefly describe the overhead system which may not be familiar to readers without railway background.

[1] Liu, Zhigang, et al. "Review of Perspectives on Pantograph-Catenary Interaction Research for High-Speed Railways Operating at 400 km/h and above." IEEE Transactions on Transportation Electrification (2023).

[2] Bruni, Stefano, et al. "Pantograph–catenary interaction: recent achievements and future research challenges." International Journal of Rail Transportation 6.2 (2018): 57-82.

[3] Y. Song, et al, "Current Collection Quality of High-speed Rail Pantograph-catenary Considering Geometry Deviation at 400 km/h and Above," in IEEE Transactions on Vehicular Technology, doi: 10.1109/TVT.2024.3404629.

(5) A more specific title is recommended for section 3, instead of a too generic one ‘materials and methods’. Actually, no materials are used in this paper.

(6) The case study prove the validation of the proposed method, but it will raise another issue. How to ensure the proposed method can give the optimal strategy of the planned interruption? No comparison with other methods or strategies is given here.

(7) Some editorial issues that may be updated:

Figure 2 is not illustrative. Please include more legends here to denote the meaning of red line and X marks.

Texts in figure 10 are too small and almost unreadable.

Author Response

Dear reviewer,

We sincerely thank you to review team for the insightful and constructive comments on this manuscript. The manuscript has been carefully revised according to the comments.

We look forward to hearing from you on the revised manuscript. In the remainder of this letter, we provide detailed answers to each of the comments.

Reviewer #1:

  • In the abstract, it is suggested to change ‘track line or the operation of …’ to ‘railway infrastructure’, as the interruption is used to maintain the infrastructure instead of just the track.

- Thanks for the comment. This has been changed by request.

  • Please include the main method used in this paper in the abstract to highlight the scientific contribution of this paper. For the current abstract, it looks like a case study report.

- Thank you for your valuable comment. The main method used in the paper was added to the abstract.

  • Again, the main theoretical method should be included in the keywords.

- Thank you. This has been added to the keywords.

  • Please distinguish the difference between the railway infrastructure and the track. Normally, there are two parts for the railway infrastructure, namely, the track and the overhead system [1-3]. It is recommended to include the following references in the introduction to briefly describe the overhead system which may not be familiar to readers without railway background.

- Thanks for the reminder. Recommended links have been included in the introduction.

[1] Liu, Zhigang, et al. "Review of Perspectives on Pantograph-Catenary Interaction Research for High-Speed Railways Operating at 400 km/h and above." IEEE Transactions on Transportation Electrification (2023).

[2] Bruni, Stefano, et al. "Pantograph–catenary interaction: recent achievements and future research challenges." International Journal of Rail Transportation 6.2 (2018): 57-82.

[3] Y. Song, et al, "Current Collection Quality of High-speed Rail Pantograph-catenary Considering Geometry Deviation at 400 km/h and Above," in IEEE Transactions on Vehicular Technology, doi: 10.1109/TVT.2024.3404629.

  • A more specific title is recommended for section 3, instead of a too generic one ‘materials and methods’. Actually, no materials are used in this paper.

- Thanks for the reminder. After considering comments from other reviewers, we changed the name of this section to Theoretical background.

  • The case study prove the validation of the proposed method, but it will raise another issue. How to ensure the proposed method can give the optimal strategy of the planned interruption? No comparison with other methods or strategies is given here.

- Thanks for the reminder. This has been added in the Discussion section where other research has been compared with our research and it has also been added in section 3.1. Planned infrastructure interruption (Figure 7).

(7) Some editorial issues that may be updated:

Figure 2 is not illustrative. Please include more legends here to denote the meaning of red line and X marks.

- Thank you. This has been added to the figures.

Texts in figure 10 are too small and almost unreadable.

  • Thank you for the valuable reminder. The Figure has been modified. For better overview, now it is figure 13.

Best regards

Authors

Reviewer 2 Report

Comments and Suggestions for Authors

Dear authors,

find below the text of my review of your manuscript. There are a lot of proposals for improvements, please consider 'em. Given the time constraints for changes to the text, consider whether it might be better to withdraw your text from the system and resubmit it later. Otherwise, you'll only have only 10 days.

with kind regards

RW

* * * * * * * * * * * * * * * * * * * * * * * * * * * * * * * * * * * * * * * * * * * * 

Thanks for the opportunity to review your manuscript entitled „Management of railway operation during the planned interruption of railway infrastructure“.

The article touches upon a serious topic, as the issue of closures on railway lines has not yet been adequately addressed. Below you will find some recommendations that I think will improve your article. These are a mix of more general recommendations on methodology as well as notes on the formality of the text.

General note on paper concept: Your manuscript is a little bit puzzling. You alternate parts of the text corresponding to the introduction and theoretical basics with the framework of calculation methodology and insights for practical application. Please, try to rearrange the text to a common structure such as: 1 Introduction; 2 Theoretical backgrounds; 3 Methodology; 4 Case study line 705; 5 Case study line 706; 6 Discussion and 7 Conclusion.

A methodology note: I appreciate that the cost to passenger carriers of overcoming disruption problems is at least partially taken into account and is not just an approach that takes into account the costs of the infrastructure manager. However, more externalities could be considered – eg. lost time by passengers due to delays and loss of connection in the hub station. And what about freight carriers? You do not consider them? There are extra costs caused by trains waiting for the end of interruption or of train divertions to other routes (see the current case with the closure of the Linz – Summerau line with divertions via ÄŒeské Velenice/Gmünd border crossing); higher personnel demand or even penalty logistics fees for late delivery of cargo to the customer in the case of time-sensitive supply chains? In general, the description of tools, methods and mathematical models (e.g. line 360 – 369; line 430-433) is vague and hampers the replicability of your approach. A more detailed description or the addition of references to third-party models used is required.

More notes:

1.         Figure 1 (line 230) is in millions of trainkm while the texts above and caption below say thousands of trainkm

2.         Table 3 (line 249) is extremely unorganized and difficult to read. I'd replace it with a diagram.

3.         The text from line 303 could be separated into independent chapters „Case study“. A map with localization of your case study lines is necessary, a few international readers of your paper have a clue where ÄŒeské BudÄ›jovice (ÄŒ.B.) town is located. Even more, I recommend having two maps: one general with the position within the whole Central Europe and another one with the ÄŒ.B. area.

4.         The line ÄŒeské BudÄ›jovice – Horní DvoÅ™ištÄ› is not a branch line (see text in line 315) – it is a mainline included in the TEN-T as you mention afterwards.

5.         The 4th transit corridor is a Czech-specific term (line 316-317) and has nothing to do with long-distance connection. To which system of corridors belongs the long-distance connection mentioned in lines 317-318? Line 706A is not included in multimodal Pan-European corridors nor RFC freight railway corridors. A link/reference will be useful here!

6.         Tables 4 and 5 (lines 313-325) – while measures for passenger transport are well described for each phase of traffic interruption in your case study, what happened to the freight transport?

7.         Line 391: where is Figure 10? (actually, the reader can find it – but two pages later!) Move it near the first occurrence in the text.

8.         For the case study in results, I would like to see a timing diagram with a detailed description of both interruptions considered, incl. inputs such as the number of trains affected

9.         Figure 11 should have been placed in the text long before this chapter (see bullet point nr. 4), please add also neighbour lines to see the possibility of train divertions.

10.    As both case studies are not equal from the viewpoint of the length of interruption, I recommend normalizing the results and comparing also eg. costs per day.

11.    There is something extra in the text in the line 785.

12.    Table 18: check the numbers – there must be a typo in the numbers for line 705

Please also fix some formal inconsistencies:

- Choose one format for presentation of costs and use it in the whole text - €1,000 vs. 1,000 €

- I would give all geographical names in the original (official) wording as they are usually presented in maps, without attempts to translate them into English (eg. "Nová Ves near ÄŒeské BudÄ›jovice" - I am pretty sure there is no map with this form of the toponymy). I recommend also keeping the diacritics in all names. Both recommendations are concerning the cartographic rule "whose language - whose name".

Being not a native speaker, I cannot evaluate well the language of the text, but I feel that it will be necessary to go through the article with an expert in railroad terms before publication.

Author Response

Dear reviewer,

We sincerely thank you to review team for the insightful and constructive comments on this manuscript. The manuscript has been carefully revised according to the comments.

We look forward to hearing from you on the revised manuscript. In the remainder of this letter, we provide detailed answers to each of the comments.

Reviewer #2:

Dear authors,

find below the text of my review of your manuscript. There are a lot of proposals for improvements, please consider 'em. Given the time constraints for changes to the text, consider whether it might be better to withdraw your text from the system and resubmit it later. Otherwise, you'll only have only 10 days.

with kind regards

RW

* * * * * * * * * * * * * * * * * * * * * * * * * * * * * * * * * * * * * * * * * * * * 

Thanks for the opportunity to review your manuscript entitled „Management of railway operation during the planned interruption of railway infrastructure“.

The article touches upon a serious topic, as the issue of closures on railway lines has not yet been adequately addressed. Below you will find some recommendations that I think will improve your article. These are a mix of more general recommendations on methodology as well as notes on the formality of the text.

General note on paper concept: Your manuscript is a little bit puzzling. You alternate parts of the text corresponding to the introduction and theoretical basics with the framework of calculation methodology and insights for practical application. Please, try to rearrange the text to a common structure such as: 1 Introduction; 2 Theoretical backgrounds; 3 Methodology; 4 Case study line 705; 5 Case study line 706; 6 Discussion and 7 Conclusion.

- Thank you for the valuable reminder. After careful consideration and consultation with other co-authors, we decided to change the Material and Methods section to Theoretical background. We have left the Methodology section and the Results section in their original state, due to the clarity of the paper and especially due to the scope of the paper. This means that we did not split the Results section into two separate sections. We hope, that you understand it . On the contrary, we have expanded the Discussion section so that everything is well interpreted and understood.

A methodology note: I appreciate that the cost to passenger carriers of overcoming disruption problems is at least partially taken into account and is not just an approach that takes into account the costs of the infrastructure manager. However, more externalities could be considered – eg. lost time by passengers due to delays and loss of connection in the hub station. And what about freight carriers? You do not consider them? There are extra costs caused by trains waiting for the end of interruption or of train divertions to other routes (see the current case with the closure of the Linz – Summerau line with divertions via ÄŒeské Velenice/Gmünd border crossing); higher personnel demand or even penalty logistics fees for late delivery of cargo to the customer in the case of time-sensitive supply chains? In general, the description of tools, methods and mathematical models (e.g. line 360 – 369; line 430-433) is vague and hampers the replicability of your approach. A more detailed description or the addition of references to third-party models used is required.

- Thanks for the comment. As part of our research, we did not consider multiple externalities, such as lost passenger time due to delays and loss of connections at a hub station. This problem was treated by the introduction of alternative bus transport, which replaced the canceled passenger trains and fast trains during infrastructure interruptions in good quality and reliability. On the contrary, in the Discussion section, we indicated a possible further direction of this research, where the lost time of passengers due to the delay or loss of the connection in the hub station will also be taken into account.

- As part of the operation of freight trains on the solved lines, all freight trains (for line 705 – 3 trains; for line 706 – 2 trains) were waiting for the end of the infrastructure interruption and were subsequently put into operation. Additional costs are caused by trains waiting for the end of the interruption or diverting trains to other routes, but in our research, which is solved on the basis of a real infrastructure interruption, freight trains did not use diversion routes and, according to available data, additional costs caused by waiting freight trains were not incurred.

- The description of tools and methods is in the Methodology section, as well as the description of models dealing with optimization and planning during emergency situations, such as infrastructure interruption.

More notes:

  1. Figure 1 (line 230) is in millions of trainkm while the texts above and caption below say thousands of trainkm

- Thank you for the reminder. This has been changed to the correct units.

  1. Table 3 (line 249) is extremely unorganized and difficult to read. I'd replace it with a diagram.

- Thanks for the comment. Table 3 has been carefully revised and edited into a more appropriate and understandable form. We did not replace the table with a diagram, as the table is clearer in this case. We hope, that you understand.

  1. The text from line 303 could be separated into independent chapters „Case study“. A map with localization of your case study lines is necessary, a few international readers of your paper have a clue where ÄŒeské BudÄ›jovice (ÄŒ.B.) town is located. Even more, I recommend having two maps: one general with the position within the whole Central Europe and another one with the ÄŒ.B. area.

- Thanks for the reminder. This has been incorporated. The map with landmarks with the location of our lines has been moved to section 3.1. and also a new map with the location of our lines within the railway network in the Czech Republic.

  1. The line ÄŒeské BudÄ›jovice – Horní DvoÅ™ištÄ› is not a branch line (see text in line 315) – it is a mainline included in the TEN-T as you mention afterwards.

- Thank you. This has been edited.

  1. The 4th transit corridor is a Czech-specific term (line 316-317) and has nothing to do with long-distance connection. To which system of corridors belongs the long-distance connection mentioned in lines 317-318? Line 706A is not included in multimodal Pan-European corridors nor RFC freight railway corridors. A link/reference will be useful here!

- Thanks for the reminder. This has been edited and clarified.

  1. Tables 4 and 5 (lines 313-325) – while measures for passenger transport are well described for each phase of traffic interruption in your case study, what happened to the freight transport?

- Thanks for the comment. This was added to tables 4 and 5. As we mentioned, all freight trains were waiting for the infrastructure interruption to end because the diversion routes were not used. We were based on a real interruption of the infrastructure.

  1. Line 391: where is Figure 10? (actually, the reader can find it – but two pages later!) Move it near the first occurrence in the text.

- Thanks for the comment. This has been incorporated and the figure has been moved to the first occurrence in the text.

  1. For the case study in results, I would like to see a timing diagram with a detailed description of both interruptions considered, incl. inputs such as the number of trains affected

- Thank you for the valuable reminder. As part of this request, tables 14 and 15 were added, which contain detailed data on both infrastructure interruptions, as well as their duration in hours, duration in days according to the individual stages of the infrastructure interruption, and the number of affected passenger and freight rail trains. Figure 14 was also added - the infrastructure interruption timetable for both lines with the marking of the individual stages of the infrastructure interruption and the specific day when the infrastructure interruption was implemented. Normal operation and interrupted operation are marked in colour in the timetable. Subsequently, both infrastructure interruptions resulting from Figure 14 are described below in the text.

  1. Figure 11 should have been placed in the text long before this chapter (see bullet point nr. 4), please add also neighbour lines to see the possibility of train divertions.

- Thanks for the reminder. The figure has been moved to section 3.1.

  1. As both case studies are not equal from the viewpoint of the length of interruption, I recommend normalizing the results and comparing also eg. costs per day.

- Thanks for the comment. This was supplemented in the Discussion section where costs and savings were evaluated. There are calculated costs per day for the operator and for the infrastructure manager.

  1. There is something extra in the text in the line 785.

- Thanks for the reminder. This has been removed.

  1. Table 18: check the numbers – there must be a typo in the numbers for line 705.

- Thank you. The values ​​in Table 20 have been modified.

Please also fix some formal inconsistencies:

- Choose one format for presentation of costs and use it in the whole text - €1,000 vs. 1,000 €

- Thank you. The format for the presentation of costs was unified to 1,000 €.

- I would give all geographical names in the original (official) wording as they are usually presented in maps, without attempts to translate them into English (eg. "Nová Ves near ÄŒeské BudÄ›jovice" - I am pretty sure there is no map with this form of the toponymy). I recommend also keeping the diacritics in all names. Both recommendations are concerning the cartographic rule "whose language - whose name".

- Thanks for the reminder. All geographical names have been edited in their original wording and with all punctuation.

Being not a native speaker, I cannot evaluate well the language of the text, but I feel that it will be necessary to go through the article with an expert in railroad terms before publication.

  • Thank you. The language of the text has been carefully checked throughout the paper.

Best regards

Authors

Round 2

Reviewer 2 Report

Comments and Suggestions for Authors

Dear authors,

There have been some positive changes in the manuscript, which makes me pleased. While reading through the updated version, I came across two things that still need to be changed to make it even better and more understandable for people unfamiliar with Czech/Slovak railways. Be aware that the readers can be from all over the world.

1) The addition of Figure 9 is good, however, for Figure 8, I would recommend adding the location of each station or stop mentioned in the text (especially in Table 4 and 5) to better illustrate how they follow each other on the line, possibly with the individual stages of the interruption marked. Instead of using a cut-off from the national map of the Railway Administration for Figure 8, I recommend preparing your own graphics in GIS (or at least in a graphic editor).

2) The adaptation of the names into Czech was fine, however not done carefully, some names were changed into Slovak instead of Czech. See:

Petríkov -> PetÅ™íkov

Hluboká pri Borovanoch -> Hluboká u Borovan

Nová Ves pri ÄŒeských BudÄ›joviciach -> Nová Ves u ÄŒeských BudÄ›jovic

Nová Ves near ÄŒeské BudÄ›jovice  -> Nová Ves u ÄŒeských BudÄ›jovic

Kaplnka -> Kaplice(?)

with regards

RW

Author Response

Dear reviewer,

We sincerely thank you to review team for the next insightful and constructive comments on this paper. The paper has been carefully revised according to the reviewer comments.

We look forward to hearing from you on the revised paper. In the remainder of this letter, we provide detailed answers to each of the comments.

Comments from the Reviewer:

Reviewer #2:

There have been some positive changes in the manuscript, which makes me pleased. While reading through the updated version, I came across two things that still need to be changed to make it even better and more understandable for people unfamiliar with Czech/Slovak railways. Be aware that the readers can be from all over the world.

  • The addition of Figure 9 is good, however, for Figure 8, I would recommend adding the location of each station or stop mentioned in the text (especially in Table 4 and 5) to better illustrate how they follow each other on the line, possibly with the individual stages of the interruption marked. Instead of using a cut-off from the national map of the Railway Administration for Figure 8, I recommend preparing your own graphics in GIS (or at least in a graphic editor).

  • Thank you for the valuable comment on this part. Figure 8 was replaced by the original figure 9 (now it is figure 8), where the position of railway lines 705 and 706 is marked as part of the infrastructure interruption in the Czech Republic.
  • Figure 9 is created in a graphic editor based on a reviewer's request. All railway stations that follow each other on the evaluated lines are marked here. The location of the railway stations is indicated along with the color marking of the individual stages of the infrastructure interruption on each line.

2) The adaptation of the names into Czech was fine, however not done carefully, some names were changed into Slovak instead of Czech. See:

Petríkov -> PetÅ™íkov

Hluboká pri Borovanoch -> Hluboká u Borovan

Nová Ves pri ÄŒeských BudÄ›joviciach -> Nová Ves u ÄŒeských BudÄ›jovic

Nová Ves near ÄŒeské BudÄ›jovice  -> Nová Ves u ÄŒeských BudÄ›jovic

Kaplnka -> Kaplice(?)

  • Thank you for the comment. The names of the railway stations were adapted to the Czech language and according to the official names. The main changes were Hluboká u Brorovan, Kaplice, PetÅ™íkov, and instead of Nová Ves u ÄŒeských BudÄ›jovic, we present the official name Nová Ves.
  • To the name of Nové Hrady, we have added the name Jakule (due to orientation on the map), because the train station Nové Hrady is located in this part, but the town of Nové Hrady is quite far from the Jakule part.

Thank you very much for your time.

Best regards

Authors